# Genome concentration limits cell growth and modulates proteome composition in *Escherichia coli*

**Jarno Mäkelä[1,2,3†], Alexandros Papagiannakis[1,2†], Wei-Hsiang Lin[1,2‡], Michael Charles Lanz[4,5‡], Skye Glenn[2,4], Matthew Swaffer[4§], Georgi K Marinov[6], Jan M Skotheim[4,5], Christine Jacobs-Wagner[1,2,4,7]\***

[1]Howard Hughes Medical Institute, Stanford University, Stanford, United States; [2]Sarafan Chemistry, Engineering, and Medicine for Human Health Institute, Stanford University, Stanford, United States; [3]Institute of Biotechnology, Helsinki Institute of Life Science, University of Helsinki, Helsinki, Finland; [4]Department of Biology, Stanford University, Stanford, United States; [5]Chan Zuckerberg Biohub, Stanford, United Kingdom; [6]Department of Genetics, Stanford University, Stanford, United States; [7]Department of Microbiology and Immunology, Stanford School of Medicine, Stanford, United States

**\*For correspondence:**
jacobs-wagner@stanford.edu

[†]These authors contributed equally to this work
[‡]These authors also contributed equally to this work

**Present address:** [§]Wellcome Centre for Cell Biology, University of Edinburgh, Edinburgh, United Kingdom

**Competing interest:** The authors declare that no competing interests exist.

## eLife Assessment

This **fundamental** work by Mäkelä et al. presents **compelling** experimental evidence supported by a theoretical model that the amount of chromosomal DNA can become limiting for the total rate of mRNA transcription and consequently protein production in the model bacterium *Escherichia coli*. The work is based on a mutant that allows inhibition of DNA replication while following growth at the single-cell level due to cell filamentation. The work significantly advances our understanding of growth and of the central dogma, and will be of considerable interest within both systems biology and microbial physiology.

**Abstract** Defining the cellular factors that drive growth rate and proteome composition is essential for understanding and manipulating cellular systems. In bacteria, ribosome concentration is known to be a constraining factor of cell growth rate, while gene concentration is usually assumed not to be limiting. Here, using single-molecule tracking, quantitative single-cell microscopy, and modeling, we show that genome dilution in *Escherichia coli* cells arrested for DNA replication limits total RNA polymerase activity within physiological cell sizes across tested nutrient conditions. This rapid-onset limitation on bulk transcription results in sub-linear scaling of total active ribosomes with cell size and sub-exponential growth. Such downstream effects on bulk translation and cell growth are near-immediately detectable in a nutrient-rich medium, but delayed in nutrient-poor conditions, presumably due to cellular buffering activities. RNA sequencing and tandem-mass-tag mass spectrometry experiments further reveal that genome dilution remodels the relative abundance of mRNAs and proteins with cell size at a global level. Altogether, our findings indicate that chromosome concentration is a limiting factor of transcription and a global modulator of the transcriptome and proteome composition in *E. coli*. Experiments in *Caulobacter crescentus* and comparison with eukaryotic cell studies identify broadly conserved DNA concentration-dependent scaling principles of gene expression.

## Introduction

Cells regulate the intracellular concentration of various proteins and macromolecules to modulate the rate of essential cellular processes, including growth. In bacteria, cell mass and volume typically double between division cycles. Proportionality between biosynthetic capacity and biomass accumulation results in exponential or near-exponential cell growth during the cell cycle (*Campos et al., 2014*; *Schaechter et al., 1958*; *Schaechter et al., 1962*; *Siegal-Gaskins and Crosson, 2008*; *Taheri-Araghi et al., 2015*; *Wang et al., 2010*). What drives exponential growth has been a longstanding question in the microbiology field (*Belliveau et al., 2021*; *Churchward et al., 1982*; *Ecker and Schaechter, 1963*; *Zhurinsky et al., 2010*). Quantitative studies on model bacteria such as *Escherichia coli* place the concentration of ribosomes and their kinetics as the principal rate-limiting factors (*Belliveau et al., 2021*; *Bosdriesz et al., 2015*; *Koch, 1988*; *Scott et al., 2014*; *Scott et al., 2010*). Most other cellular components essential for growth are estimated to be at least an order of magnitude above the level required for proper enzymatic reactions (*Belliveau et al., 2021*), indicating that they are well in excess in terms of metabolic concentrations. Thus, translation is generally seen as the rate-governing process for cellular growth. While the translocation rate of ribosomes poses an inherent limit on the growth rate of the cell, protein concentrations are predominantly set transcriptionally at the promoter level, with tight coordination between transcription and translation (*Balakrishnan et al., 2022*).

Whereas the importance of ribosome concentration in growth rate determination has been extensively studied, a potential role for genome concentration has received less attention. An early population study on an *E. coli* thymine auxotroph proposed that global transcription is not limited by the concentration of the genome but is instead constrained by the availability of RNA polymerases (RNAPs) (*Churchward et al., 1982*). However, the potential impact of DNA concentration on determining the growth rate of *E. coli* or other bacteria has, to our knowledge, not been formally tested. Interestingly, *E. coli* and *Bacillus subtilis* have been shown to display small but reproducible deviations from exponential growth during the division cycle (*Kar et al., 2021*; *Nordholt et al., 2020*), with the growth rate increasing after the initiation of DNA replication under some conditions. Furthermore, at the population level, these organisms initiate DNA replication at a fixed cell volume (mass) per chromosomal origin of replication (*oriC*) across a wide range of nutrient and genetic conditions (*Donachie, 1968*; *Govers et al., 2024*; *Si et al., 2017*; *Zheng et al., 2016*), suggesting that DNA concentration is an important physiological parameter for these bacteria. In eukaryotes where genome concentration is also tightly controlled (*Ginzberg et al., 2015*; *Turner et al., 2012*), a change in DNA-to-cell-volume ratio has recently been demonstrated to remodel the proteome and promote cellular senescence (*Crozier et al., 2023*; *Foy et al., 2023*; *Lanz et al., 2024*; *Lanz et al., 2022*; *Manohar et al., 2023*; *Neurohr et al., 2019*; *Wilson et al., 2023*).

In this study, we combined single-cell and single-molecule microscopy experiments with tandem-mass-tag (TMT)-mass spectrometry (MS), RNA sequencing (RNA-seq), and modeling to investigate the potential physiological role of genome concentration in cell growth and proteome composition in *E. coli*.

## Results

### Growth rate correlates with the genome copy number

To examine the potential effect of DNA content on the growth rate of *E. coli*, we used two CRISPR interference (CRISPRi) strains with arabinose-inducible control of expression of dCas9 (*Li et al., 2016*; *Si et al., 2017*). One strain expressed a single-guide RNA (sgRNA) against *oriC* where sequestration by dCas9 binding prevents the initiation of DNA replication to produce cells with a single copy of the chromosome after already initiated DNA replication rounds are completed and cells undergo reductive division (*Si et al., 2017*). These cells, referred to as '1N cells' below, grew into filaments as a block in DNA replication prevents cell division, but not cell growth, from occurring (*Figure 1A*; *Carl, 1970*; *Si et al., 2017*; *Withers and Bernander, 1998*). The second CRISPRi strain, which served as a comparison, expressed an sgRNA against the cell division protein FtsZ. FtsZ depletion blocks cell division while allowing DNA replication to proceed (*Addinall et al., 1996*; *Li et al., 2016*). Ongoing growth resulted in filamenting cells with multiple replicating chromosomes, hereafter referred to as 'multi-N cells' (*Figure 1A*). For both strains, we used time-lapse microscopy to monitor growth at the single-cell level at 37°C in M9 minimal medium supplemented with glycerol, casamino acids, and

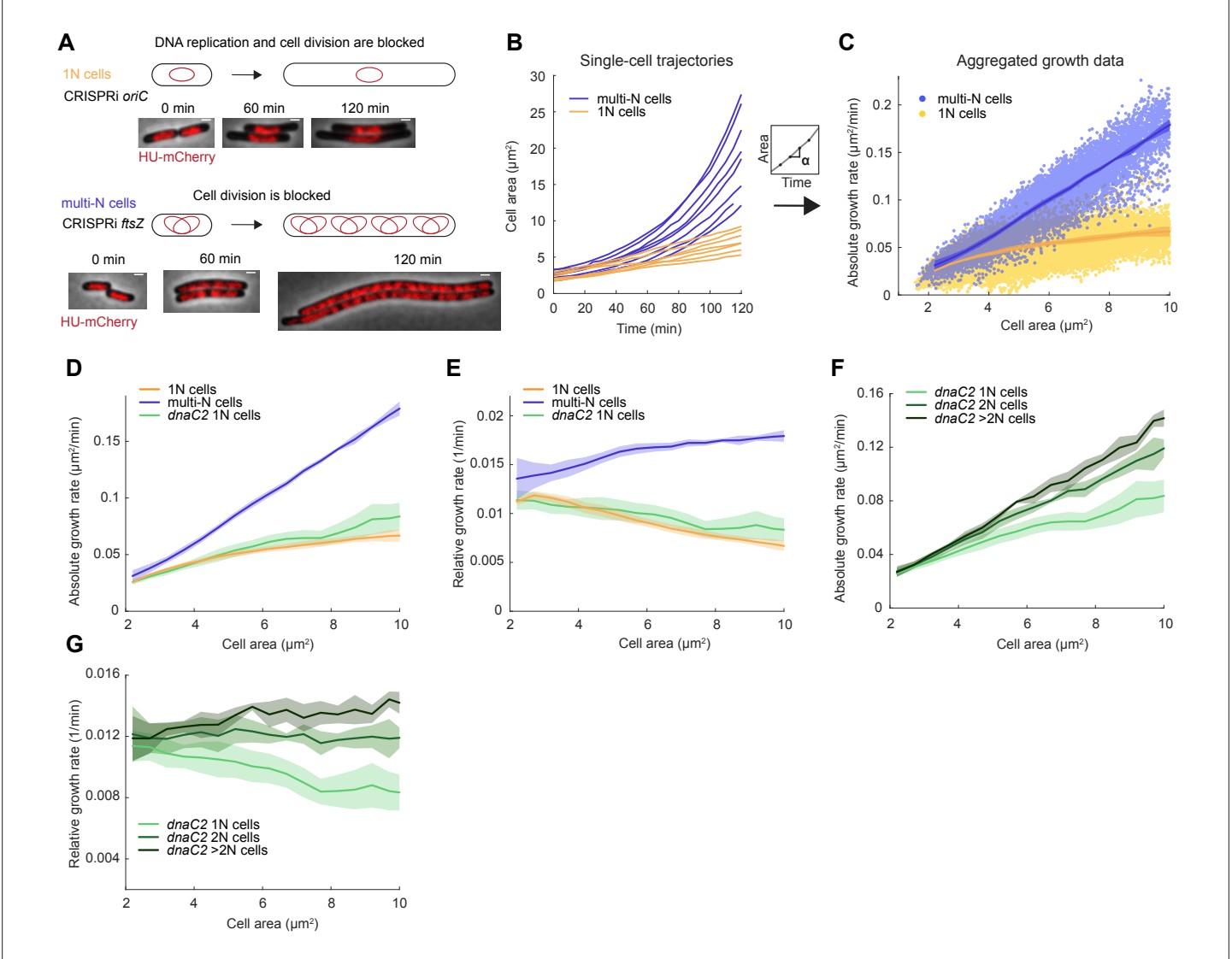

**Figure 1.** Growth rate and genome copy number in *E. coli* growing in M9glyCAAT. (**A**) Illustration of 1N (CRISPR interference [CRISPRi] *oriC*, CJW7457) and multi-N (CRISPRi *ftsZ*, CJW7576) cells with different numbers of chromosomes along with representative microscopy images at different time points following CRISPRi induction. Scale bars: 1 μm. (**B**) Plot showing representative single-cell trajectories of cell area as a function of time for the CRISPRi strains following a block in DNA replication and/or cell division. (**C**) Plot showing the absolute growth rate as a function of cell area for 1N (32735 datapoints from 1568 cells) and multi-N cells (14,006 datapoints from 916 cells) in M9glyCAAT. Lines and shaded areas denote mean ± SD from three experiments. This also applies to the panels below. (**D**) Absolute and (**E**) relative growth rate in 1N (32735 datapoints from 1568 cells, CJW7457), multi-N (14,006 datapoints from 916 cells, CJW7576), and *dnaC2* 1N (13,933 datapoints from 1043 cells, CJW7374) cells as a function of cell area in M9glyCAAT. (**F**) Absolute and (**G**) relative growth rate in 1N (13,933 datapoints from 1043 cells), 2N (6265 datapoints from 295 cells), and >2N (2116 datapoints from 95 cells) *dnaC2* (CJW7374) cells as a function of cell area in M9glyCAAT.

The online version of this article includes the following figure supplement(s) for figure 1:

**Figure supplement 1.** The relative growth rate of wild-type (WT) (CJW7339) cells in M9glyCAAT grown after placing them on an agarose pad.

**Figure supplement 2.** Characterization of ploidy in CRISPR interference (CRISPRi) *oriC* cells.

**Figure supplement 3.** Effects of cell area overestimation on the relative growth rate calculation.

**Figure supplement 4.** Relationship between growth rate and cell area in cell types of different ploidy.

**Figure supplement 5.** Validation of stable growth under microscope observation and absolute growth rate determination of ppGpp⁰ and ΔrecA cells.

**Figure supplement 6.** Characterization of ploidy in *dnaC2* cells.

**Figure supplement 7.** Relationships between growth rate and cell volume across cell types of different ploidy in M9glyCAAT.

**Figure supplement 8.** DNA-dependent growth in *C. crescentus*.

thiamine (M9glyCAAT). Cell area ($A$) was automatically detected from phase-contrast images using a deep convolutional network (**Wiktor et al., 2021**), and the absolute growth rate ($\frac{dA}{dt}$) was determined by calculating the difference in cell area between frames. The relative growth rate ($\frac{1}{A}\frac{dA}{dt}$), which is constant for exponential growth, was calculated by dividing the absolute growth rate by the cell area. We used wild-type (WT) cells to verify that the transition from liquid cultures to agarose pads led to stable growth from the start of image acquisition (**Figure 1—figure supplement 1**).

As the induced CRISPRi *oriC* phenotype is not fully penetrant, we limited our analysis to 1N cells that contained a single DNA object (nucleoid) labeled by a mCherry fusion to the nucleoid-binding protein HupA (referred to as HU below). To confirm this 1N chromosome designation, we used a CRISPRi *oriC* strain that expresses HU-CFP and carries an *oriC*-proximal *parS* site labeled with ParB-mCherry (**Figure 1—figure supplement 2**), used here to determine the number of nucleoids and chromosomal origins per cell. We found that 96 ± 1% (mean ± standard deviation, SD, three biological replicates) of cells (n=3378) with a single HU-labeled nucleoid contained no more than one ParB-mCherry focus, indicative of a single *oriC*.

Using this methodology, we observed a significant difference in growth rate between 1N and multi-N cells as shown in representative single-cell growth trajectories (**Figure 1B**) and in aggregated absolute growth rate measurements (**Figure 1C**). In multi-N cells, the absolute growth rate rapidly increased with cell area. In 1N cells, the absolute growth rate only moderately increased with cell area, approaching an apparent plateau at large cell sizes (**Figure 1C**). As an independent validation, we used an orthogonal system to block DNA replication using the temperature-sensitive mutant *dnaC2*, which encodes a deficient DNA helicase loader at the restrictive temperature of 37°C (**Carl, 1970**; **Withers and Bernander, 1998**). We observed that the relationship between absolute growth rate and cell area in *dnaC2* cells with a single nucleoid was similar to that of 1N cells produced by the CRISPRi *oriC* system (**Figure 1D**). This sub-exponential growth in 1N and *dnaC2* cells resulted in a relative growth rate that decreased with cell area (**Figure 1E**). For multi-N cells, the relative growth rate was not perfectly constant but appeared to increase somewhat with cell area (**Figure 1E**). It is unclear whether this slight increase is biologically meaningful, as simulations show that a small inaccuracy in cell size from cell segmentation can produce the appearance of super-exponential growth (**Figure 1—figure supplement 3**). Regardless, and most importantly, the multi-N cells grew identically to WT within the same cell size range while 1N cells grew significantly slower (**Figure 1—figure supplement 4**).

The striking divergence in growth between 1N and multi-N cells of the same size suggested that DNA concentration can affect growth rate. The difference in growth rate between 1N and multi-N cells was already apparent in the physiological range of cell sizes when compared to WT cells (**Figure 1—figure supplement 4**), suggesting that growth rate reduction occurs soon after DNA replication fails to initiate. We confirmed that the slower growth of 1N cells did not depend on the time that cells spent on agarose pads (**Figure 1—figure supplement 5A**). We also ruled out that the growth reduction was due to an induction of the SOS response or to an increased level in the nucleotide alarmone (p)ppGpp, as inactivation of either stress pathway (through deletion of *recA* or *spoT/relA*, respectively) in 1N cells made little to no difference to their growth rate (**Figure 1—figure supplement 5B**).

We noticed that, even at the restrictive temperature, the *dnaC2* strain produced a sizeable fraction of cells with more than one HU-mCherry-labeled nucleoid (**Figure 1—figure supplement 6A and B**), indicating that the temperate-sensitive effect on DNA replication is not fully penetrant. We took advantage of this phenotypic 'leakiness' to measure the growth rate of cells with different numbers of nucleoids (and thus chromosomes) within the *dnaC2* population. We observed a notable difference in growth rate between cells of 1, 2, and >2 nucleoids in the population, with each additional nucleoid contributing to higher cellular growth at a given cell size (**Figure 1F and G**). This finding is consistent with DNA-limited growth in which cellular growth rate increases with genome concentration. We obtained similar results when we calculated absolute and relative growth rates based on extracted cell volumes instead of areas (**Figure 1—figure supplement 7A–F**), as cell width remained largely constant during cell filamentation (**Figure 1—figure supplement 7G**).

A growth rate dependency on genome concentration is unlikely to be a particularity of *E. coli*, as we also observed a divergence in absolute and relative growth rates with increasing cell area between 1N and multi-N cells of *C. crescentus* (**Figure 1—figure supplement 8A and B**). We generated filamenting 1N and multi-N *C. crescentus* cells by depleting the DNA replication initiation factor DnaA (**Gorbatyuk and Marczynski, 2001**) and the cell division protein FtsZ (**Wang et al., 2001**),

respectively. We confirmed the 1N vs. multi-N designation by visualizing the number of chromosomal origins of replication (one vs. multiple) per cell using the *parS*/ParB-eCFP labeling system (*Figure 1—figure supplement 8C*).

## The concentration of ribosomal proteins remains relatively constant in genome-diluted *E. coli* cells

Ribosome content is often proposed to explain the exponential growth of biomass in bacteria, with growth rate being directly proportional to ribosome concentration (*Bremer and Dennis, 2008*; *Ecker and Schaechter, 1963*; *Scott et al., 2014*; *Scott et al., 2010*). Therefore, we first quantified the fluorescence concentration of a monomeric superfolder green fluorescent protein (msfGFP) fusion to the ribosomal protein RpsB (expressed from the native chromosomal locus) in 1N and multi-N cells in M9glyCAAT as a function of cell area. We found it to be almost identical between the two CRISPRi strains and relatively constant across cell areas, regardless of DNA content (*Figure 2A*).

To exclude the possibility that the msfGFP tag altered the synthesis of RpsB or that this protein behaved differently from other ribosomal proteins, we adapted a TMT MS method recently developed to examine cell size-dependent proteome scaling in yeast and human cells (*Lanz et al., 2022*). Note that, for the CRISPRi *oriC* strain, a minority (~10–15%) of cells have more than one nucleoid. These cells were excluded from the analysis of our single-cell microscopy experiments. However, this could not be done for the TMT-MS experiments, which provide population-level measurements. Therefore, for this TMT-MS section, we will refer to the CRISPRi *oriC* cell population as '1N-rich' cells, instead of only '1N' cells. Using the TMT-MS approach, we found that the relative concentration of all (54) high-abundance ribosomal proteins (including untagged RpsB) remained approximately constant across all sizes of 1N-rich cells, and was similar between 1N-rich and multi-N cells (*Figure 2B*). Only the relative concentration of the ribosomal protein L31B, a stationary phase paralog of the more prevalent exponential phase ribosomal protein L31A (*Lilleorg et al., 2019*), significantly decreased in 1N cells (*Supplementary file 1*). Thus, the concentration of ribosomal proteins does not explain the difference in growth rate between cells with different ploidy.

## The fraction of active ribosomes is reduced in genome-diluted cells

To more specifically probe the translational activity of ribosomes in 1N cells, we performed single-molecule tracking in live cells growing in M9glyCAAT. Ribosomes are expected to exhibit at least two different dynamic states: slow mobility when active (i.e. engaged in translation on the mRNA, often in polyribosome form), and faster mobility when inactive ribosomes (or ribosomal subunits) are diffusing in the cytoplasm (*Mohapatra and Weisshaar, 2018*; *Sanamrad et al., 2014*). To track ribosomes, we introduced a HaloTag fusion to RpsB (through genetic modification at the endogenous chromosomal locus) and labeled the HaloTag using the membrane-permeable Janelia Fluor 549 (JF549) fluorescent dye (*Grimm et al., 2015*). We quantified the apparent diffusion coefficient ($D_a$) of single-molecule tracks in WT cells, as well as in 1N and multi-N cells at multiple time points following CRISPRi induction (*Figure 2C*). We found that the distribution of $D_a$ in multi-N cells of all sizes (~2–10 μm$^2$) was similar to that in WT cells despite the considerable differences in cell sizes. In contrast, 1N cells displayed distributions clearly distinct from WT and multi-N cells, gradually shifting toward faster mobilities (higher $D_a$) with increasing cell size. This shift suggests that ribosome activity is altered in 1N cells.

Gaussian fitting of the $D_a$ logarithmic data in WT cells revealed two predominant dynamic states of ribosomes: a slow-diffusing and a fast-diffusing state, representing 77 ± 1% (mean ± standard error of the mean [SEM]) and 20 ± 1% of the ribosome population, respectively (*Figure 2D*). In addition, we observed a small fraction (3.2 ± 0.5%) of faster-moving molecules with $D_a$ expected for freely diffusing proteins (*Banaz et al., 2019*; *Elowitz et al., 1999*), likely indicative of a small pool of free RpsB-HaloTag proteins (i.e. not assembled into ribosomes). To confirm that the slow-diffusing fraction corresponded to translationally active ribosomes, we showed that this fraction nearly vanished (down to 1.10 ± 0.02%) when cells were depleted of mRNAs following 30 min treatment with the transcription inhibitor rifampicin (*Figure 2—figure supplement 1*). The estimated fraction (~77%) of active ribosomes in untreated cells was in good agreement with previous single-molecule and biochemical studies under similar growth conditions (*Forchhammer and Lindahl, 1971*; *Mohapatra and Weisshaar, 2018*; *Sanamrad et al., 2014*).

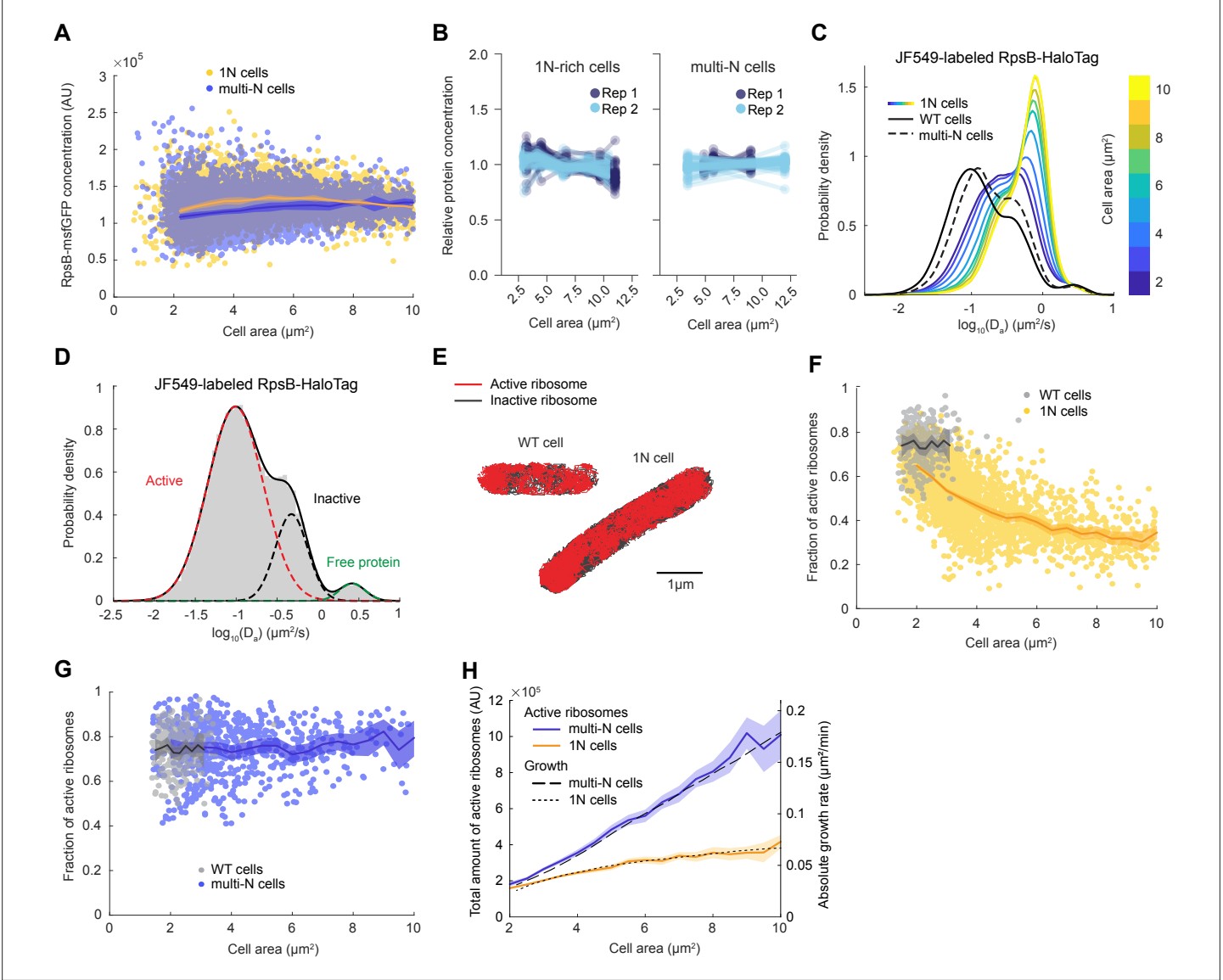

**Figure 2.** Lower ribosome activity explains the reduced growth rate of 1N cells growing in M9glyCAAT. (**A**) RpsB-msfGFP fluorescence concentration in 1N (6542 cells, CJW7478) and multi-N (10,537 cells, CJW7564) cells as a function of cell area. Lines and shaded areas denote mean ± SD from three experiments. (**B**) Relative protein concentration of different ribosomal proteins in 1N (SJ_XTL676) and multi-N (SJ_XTL229) cells by tandem-mass-tag (TMT)-mass spectrometry (MS). 1N-rich cells were collected 0, 120, 180, 240, and 300 min after addition of 0.2% arabinose, while multi-N cells were collected after 0, 60, and 120 min of induction. Blue and cyan represent two independent experiments. Only proteins with at least four peptide measurements are plotted. (**C**) Apparent diffusion coefficients ($D_a$) of JF549-labeled RspB-HaloTag in wild-type (WT) (32,410 tracks from 771 cells, CJW7528), 1N (848,367 tracks from 2478 cells, CJW7529), and multi-N cells (107,095 tracks from 1139 cells, CJW7530). Only tracks of length ≥9 displacements are included. 1N cells are color-binned according to their cell area while multi-N cells contain aggregated data for ~2–10 μm² cell areas. (**D**) $D_a$ in WT cells fitted by a three-state Gaussian mixture model (GMM): 77 ± 1%, 20 ± 1%, and 3.2 ± 0.5% (± standard error of the mean [SEM]) of the ribosome population, from the slowest moving to the fastest moving (32,410 tracks from 771 cells). (**E**) Example WT and 1N cells where active (red, slow-moving) and inactive (gray, fast-moving) ribosomes are classified according to the GMM. (**F**) Active (slow-moving) ribosome fraction in individual WT (237 cells) and 1N (2453 cells) cells as a function of cell area. Only cells with ≥50 tracks are included. Lines and shaded areas denote mean and 95% confidence interval (CI) of the mean from bootstrapping. (**G**) Same as (**F**) but for WT (237 cells) and multi-N (683 cells) cells. (**H**) Absolute growth rate of 1N and multi-N cells (*Figure 1C*) as a function of cell area was overlaid with the total active ribosome amount (calculated from **A, F, and G**). Lines and shaded areas denote mean and 95% CI of the mean from bootstrapping. All microscopy data are from three biological replicates. msfGFP, monomeric superfolder green fluorescent protein.

The online version of this article includes the following figure supplement(s) for figure 2:

**Figure supplement 1.** Diffusive characteristics of labeled ribosomes in rifampicin-treated wild-type (WT) cells.

**Figure supplement 2.** Diffusive characteristics of labeled ribosomes in 1N cells as a function of cell area.

Upon fitting the $D_a$ values of ribosomes in WT and 1N cells (*Figure 2E*), we observed a significant reduction in the slow-diffusing ribosome population in 1N cells of increasing area (*Figure 2—figure supplement 2*). Quantification of the active (slow-diffusing) ribosome fraction per cell revealed that 1N cells have overall lower ribosome activity than WT cells, and that ribosome activity decreases monotonically with increasing cell area (*Figure 2F*). In contrast, ribosome activity in multi-N cells remained the same as in WT across different cell sizes (*Figure 2G*).

To estimate the total number of active ribosomes per cell, we multiplied the total amount of ribosomes by the fraction of active ribosomes and plotted the result as a function of cell area (*Figure 2H*). We found that the difference in the total number of active ribosomes between 1N and multi-N cells matches the observed difference in growth rate (*Figure 2H*), indicating that cell growth rate is directly proportional to the increase in total active ribosomes. Altogether, the results are consistent with the hypothesis that DNA limitation decreases total ribosome activity, which, in turn, reduces the growth rate.

## Genome dilution reduces the activity of RNAPs

We reasoned that the observed changes in ribosome activity in 1N cells may reflect the available pool of transcripts. If true, we would expect the total activity of RNAPs to be reduced in 1N cells. The total activity of RNAPs in cells is determined by the concentration of RNAPs multiplied by the fraction of active RNAPs. Therefore, we first determined whether RNAP concentration was lower in 1N cells relative to multi-N cells by quantifying the fluorescence intensity of a functional fusion of YFP to the RNAP β' subunit (encoded by *rpoC*) expressed from its native chromosomal locus. As expected, RNAP concentration remained constant in multi-N cells (*Figure 3A*). In 1N cells, the RNAP concentration increased with cell size (*Figure 3A*), the opposite of what would be expected to explain the growth rate defect. We confirmed this increasing trend in concentration for other protein subunits of the core RNAP and the primary sigma factor σ⁷⁰ (encoded by *rpoD*) using TMT-MS (*Figure 3B*), clearly demonstrating that the abundance of RNAPs was not the limiting factor.

To quantify RNAP activity in 1N and multi-N cells, we performed single-molecule tracking in live cells using a functional fusion of HaloTag to the β' protein subunit RpoC labeled with the JF549 dye. As expected, the $D_a$ values of RpoC-HaloTag in multi-N cells were distributed similarly to those in WT cells (*Figure 3C*). In contrast, the distribution in 1N cells changed gradually toward higher $D_a$ values (faster mobility) with increasing cell size (*Figure 3C*). As with ribosomes, RNAPs primarily exhibited two major states of diffusivity (*Figure 3D*): a slower-diffusing fraction (49 ± 4%; mean ± standard error of the mean [SEM]) and a faster-diffusing fraction (49 ± 4%), likely representing transcriptionally active RNAPs and inactive, diffusing RNAPs, respectively. A small fraction of RpoC-HaloTag (2 ± 0.1%) diffused very fast, with $D_a$ values expected for free proteins, suggesting that it reflects the few β' proteins not assembled into the RNAP core complex. Using rifampicin treatment, we confirmed that the slowest state corresponds to RNAPs actively engaged in transcription (*Figure 3—figure supplement 1*). In these rifampicin-treated cells, the slow-diffusing fraction was reduced to 13 ± 4%. Rifampicin does not prevent promoter binding or open complex formation and instead blocks transcription elongation following the synthesis of 3-nucleotide-long RNAs (*Campbell et al., 2001*). Thus, the observation that slow-moving RNAPs did not completely disappear after rifampicin treatment is consistent with the mechanism of action of the drug, leaving a fraction of RNAPs bound at promoter sites.

Unlike in WT and multi-N cells, the fraction of active RNAPs in 1N cells decreased monotonically with increasing cell area (*Figure 3F and G*). However, because the RNAP concentration simultaneously increased in 1N cells, it remained possible that the total amount of active RNAP, which is the relevant metric of transcription activity, remained equal to that of multi-N cells. By calculating the total amount of active RNAPs, we showed that the decrease in the active fraction in 1N cells was not the mere result of the increase in RNAP concentration. Indeed, the total amount of active RNAPs hardly increased with cell size in 1N cells whereas it increased proportionally with cell size in both multi-N and WT cells (*Figure 3H*).

A recent study has shown that the intracellular concentration of Rsd, the anti-sigma factor of σ⁷⁰, increases in WT cells under slower growth conditions, causing a reduction in global mRNA synthesis (*Balakrishnan et al., 2022*). Therefore, we verified that the concentration of Rsd remains approximately constant in both 1N-rich and multi-N cells based on our TMT-MS data (*Figure 3—figure*

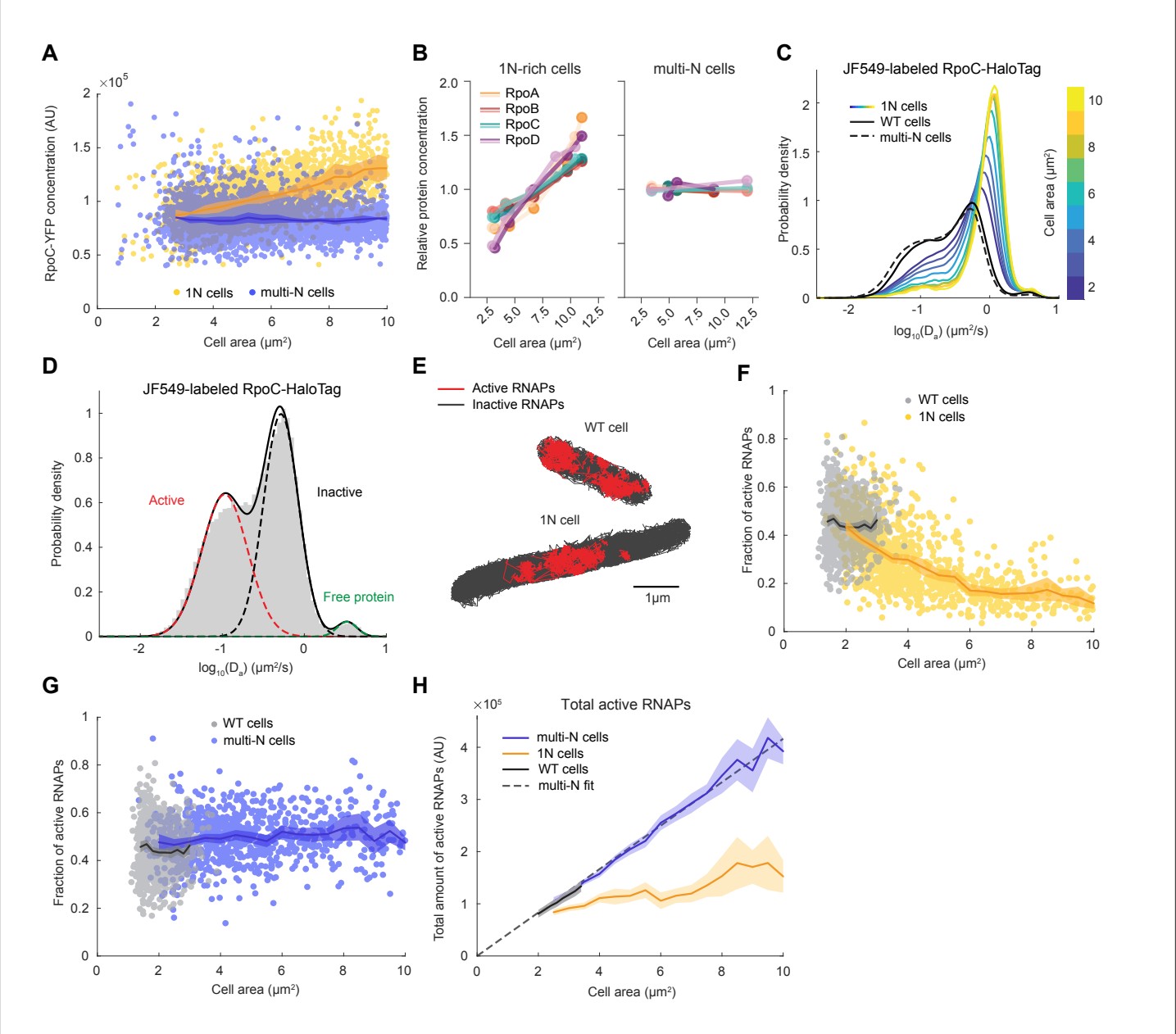

**Figure 3.** RNA polymerase (RNAP) activity is reduced in 1N cells growing in M9glyCAAT. (**A**) RpoC-YFP fluorescence concentration in 1N (3580 cells, CJW7477) and multi-N (5554 cells, CJW7563) cells as a function of cell area. Lines and shaded areas denote mean ± SD from three experiments. (**B**) Relative protein concentration of core RNAP subunits and σ⁷⁰ in 1N-rich (SJ_XTL676) and multi-N (SJ_XTL229) cells by tandem-mass-tag (TMT)-mass spectrometry (MS). 1N-rich cells were collected 0, 120, 180, 240, and 300 min after addition of 0.2% L-arabinose, while multi-N cells were collected after 0, 60, and 120 min of induction. (**C**) Apparent diffusion coefficients of JF549-labeled RpoC-HaloTag in wild-type (WT) (91,280 tracks from 1000 cells, CJW7519), 1N (175,884 tracks from 1219 cells, CJW7520) and multi-N cells (186,951 tracks from 1040 cells, CJW7527). Only tracks of length ≥9 displacements are included. 1N cells are binned according to cell area while multi-N cells contain aggregated data for ~2–15 μm² cell areas. (**D**) $D_a$ in WT cells fitted by a three-state Gaussian mixture model (GMM): 49 ± 4%, 49 ± 4%, and 2 ± 0.1% (± standard error of the mean [SEM]) of the RNAP population, from the slowest moving to the fastest moving (91,280 tracks from 1000 cells). (**E**) Example WT and 1N cells where active (red, slow-moving) and inactive (gray, fast-moving) RNAPs are classified according to the GMM. (**F**) Active RNAP fraction in individual WT (854 cells) and 1N (1024 cells) cells as a function of cell area. Only cells with at least 50 tracks are included. Lines and shaded areas denote mean ±95% CI of the mean from bootstrapping (three experiments). (**G**) Same as (**F**) but for WT (854 cells) and multi-N (924 cells) cells. (**H**) Total amount of active RNAP in WT, 1N, and multi-N cells as a function of cell area (calculated from **A, F,** and **G**). Also shown is a linear fit to multi-N data ($f(x) = 4.16 \cdot 10^4 \cdot x$, $R^2$ 0.98). Lines and shaded areas denote mean and 95% CI of the mean from bootstrapping. All microscopy data are from three biological replicates.

The online version of this article includes the following figure supplement(s) for figure 3:

*Figure 3 continued on next page*

**Figure supplement 1.** Diffusive characteristics of labeled RNA polymerases (RNAPs) in rifampicin-treated cells.

**Figure supplement 2.** Determination of the relative Rsd concentration in 1N-rich and multi-N cells as a function of cell area.

*supplement 2*), eliminating Rsd as a possible source of reduced RNAP activity in 1N cells. Instead, our data supports the notion that substrate (DNA) limitation leads to a reduced transcription rate, which reduces the pool of transcripts available for ribosomes.

## Chromosome dilution reduces the concentration of transcripts

To test the idea that genome dilution affects growth rate through transcript limitation, we performed live-cell staining with SYTO RNASelect, a fluorogenic RNA-specific dye (*Wu et al., 2020*). This dye has been proposed to preferentially bind mRNAs based on the observed decay of intracellular RNASelect signal in *E. coli* during rifampicin treatment (*Bakshi et al., 2014*), which causes mRNA depletion. However, a recent study has shown that the levels of ribosomal RNAs (rRNAs) also decrease in rifampicin-treated cells (*Hamouche et al., 2021*), though at a slower rate than mRNAs. Therefore, to complement the RNASelect staining experiments and examine the potential effect of genome dilution specifically on rRNAs, we also carried out fluorescence in situ hybridization (FISH) microscopy on fixed cells using EUB338-Cy3, a DNA probe complementary to an exposed region in the 16S rRNA (*Amann et al., 1990*). For both experiments, we mixed 1N cells with multi-N cells of similar size ranges prior to incubation with RNASelect or EUB338-Cy3 to mitigate variability in staining. We next imaged the mixed populations and distinguished 1N cells from multi-N cells by examining the difference in nucleoid number (one vs. multiple) per cell using HU-mCherry or DAPI as a DNA marker (*Figure 4A*). Single cells were sampled to ensure that the cell area distributions of the two populations matched (*Figure 4—figure supplement 1*).

Comparison between the two sampled populations revealed a reduced concentration of RNASelect signal by ~50% in 1N cells relative to multi-N cells for a cell size range of 4–10 μm$^2$ (*Figure 4B and C*). For a similar cell area range, the EUB338 signal concentration was reduced by only ~5%. Furthermore, the RNASelect concentration ratio between 1N and multi-N cells displayed a rapid exponential decay with increasing cell area, whereas the decrease in EUB338 concentration ratio was considerably slower (*Figure 4D*).

To verify that the decrease in RNASelect signal in 1N cells was not caused by a global change in membrane permeability to small molecules, we performed similar live-cell staining experiments with the HaloTag dye JF549 in CRISPRi strains expressing RpoC-HaloTag (*Figure 4—figure supplement 2A*). We matched the cell distributions between 1N and multi-N cells for fair comparison (*Figure 4—figure supplement 2B*). Because RpoC concentration increases with cell size in 1N cells relative to multi-N cells (*Figure 3A and B*), we expected a similar increase in the ratio of JF549 signal between these two cell types if the membrane permeability to small molecules remained unchanged. This is indeed what we observed (*Figure 4—figure supplement 2C–E*). In parallel, to examine the ability of our rRNA FISH method to detect a reduction in 16S rRNA concentrations, we compared the EUB338 staining of WT cells (MG1655) growing in M9 glycerol with or without casamino acids and thiamine (M9glyCAAT vs. M9gly), which results in a difference in growth rate of ~40% (*Govers et al., 2024*) due to the expected lower concentration of ribosomes and thus 16S rRNAs in nutrient-poor media. Consistent with this expectation, we found that the EUB338 concentration signal was reduced by ~50% in M9gly relative to M9glyCAAT (*Figure 4—figure supplement 3*). Given these validations, our results in *Figure 4* suggest that the RNASelect signal primarily reflects the bulk of mRNAs, and that the concentration of mRNAs decreases more rapidly than that of rRNAs upon genome dilution.

## DNA dilution can result in sub-exponential growth through mRNA limitation

In a previous theoretical study, *Lin and Amir, 2018* considered distinct scenarios for gene expression. Their model predicted that if DNA and mRNAs are in excess, cells will display exponential growth. On the other hand, cells will adopt linear growth if DNA and mRNAs become limiting. Our experiments showed that 1N cells indeed converge toward linear growth (toward slope 0 in *Figure 1C*), though the complete transition to linear growth required a large decrease in DNA concentration. To quantitatively

examine this transition from exponential to linear growth through genome dilution, we developed two deterministic ordinary differential equation (ODE) models of the flow of genetic information that include parameters for the fractions of active RNAPs and ribosomes. In these models, the dynamics of mRNA ($X$) and protein ($Y$) numbers in the cell are described by

$$\frac{dX}{dt} = r_1 \alpha_{RNAP}\left(X, Y\right) Y - \delta X$$

$$\frac{dY}{dt} = r_2 \alpha_{ribo}\left(X, Y\right) Y$$

where $r_1$ is the bulk transcription rate normalized by the total protein number, $r_2$ is the bulk translation rate normalized by the total protein number, and $\delta$ is the mRNA degradation rate. The quantities $\alpha_{RNAP}\left(X, Y\right)$ and $\alpha_{ribo}\left(X, Y\right)$ are the fractions of active RNAPs and ribosomes expressed as a percentage of the total RNAPs and ribosomes, respectively. For simplicity, we assumed that protein degradation is negligible and that the cell volume and the number of rRNAs grow proportional to protein $Y$ (*Balakrishnan et al., 2022*; *Lin and Amir, 2018*). As a result, the rate of protein increase $\frac{dY}{dt}$ corresponds to the absolute growth rate and the relative protein increase rate $\frac{1}{Y}\frac{dY}{dt}$ corresponds to the relative growth rate. For detailed description and estimation of the model parameters, see *Supplementary file 2* and Appendices 1 and 2.

Based on the function form of $\alpha_{RNAP}\left(X, Y\right)$, we consider two ODE model variants. In model A, we assumed that DNA is a limiting factor while RNAPs are not. In model B, both DNA and RNAPs were considered as growth-limiting factors. In both models, higher DNA concentration increases the probability that an RNAP will encounter and bind to a promoter. In model terms, $\alpha_{RNAP}$ (as well as the downstream transcription rate) increases with DNA concentration. In model A, we examined the effect of DNA limitation with minimal mathematical complexity by assuming that the proteome does not change (see Materials and methods). In model B, we considered RNAP kinetics (with three different RNAP states: free, promoter-bound, and transcribing) based on the law of mass action (see Materials and methods and Appendix 3) and took into consideration the experimentally observed increase in RNAP concentration in 1N cells (*Figure 3A and B*). For both models, $\alpha_{RNAP}$ depended on DNA concentration.

We used these models to perform simulations and compared the results to our measurements, starting with parameter values extracted or estimated from the *E. coli* literature (*Supplementary file 3*). In 1N cells, the DNA amount was fixed to one genome while it scaled with cell volume in multi-N cells. The parameters were then optimized to fit six experimental datasets simultaneously: cell growth rate, the fraction of active RNAPs, and the fraction of active ribosomes in both 1N and multi-N cells (see Materials and methods and Appendix 4).

As shown in *Figure 5A–D* (model A) and *Figure 5—figure supplement 1* (model B), both models performed similarly after parameter optimization. While the model curves (solid lines) did not perfectly match the average behavior of our experimental results (open squares), they displayed similar trends and fell within the variance of the single-cell data (dots). The models showed that multi-N cells (blue) display balanced exponential growth while the 1N cells (yellow) exhibit sub-exponential growth (*Figure 5A* and *Figure 5—figure supplement 1A*), consistent with experiments. At the same time, both models recapitulated the observed experimental trends in active fractions of both ribosomes and RNAPs, which remained constant in multi-N cells while decaying gradually with DNA concentration in 1N cells (*Figure 5B–D* and *Figure 5—figure supplement 1B–D*).

The simulation results of 1N cells suggest the following cascade of events when DNA is limiting. Lower DNA concentration results in fewer substrates for RNAPs, which reduces the transcription rate. This results in a decrease in mRNA concentration. As mRNAs become limiting, the fraction of ribosomes engaged in translation decreases. This, in turn, decreases the rate of bulk protein synthesis, which decreases the relative growth rate. The greater the DNA dilution (through cell growth), the more severe the downstream effects become, explaining the decay in relative growth rate in 1N cells (*Figure 5E*).

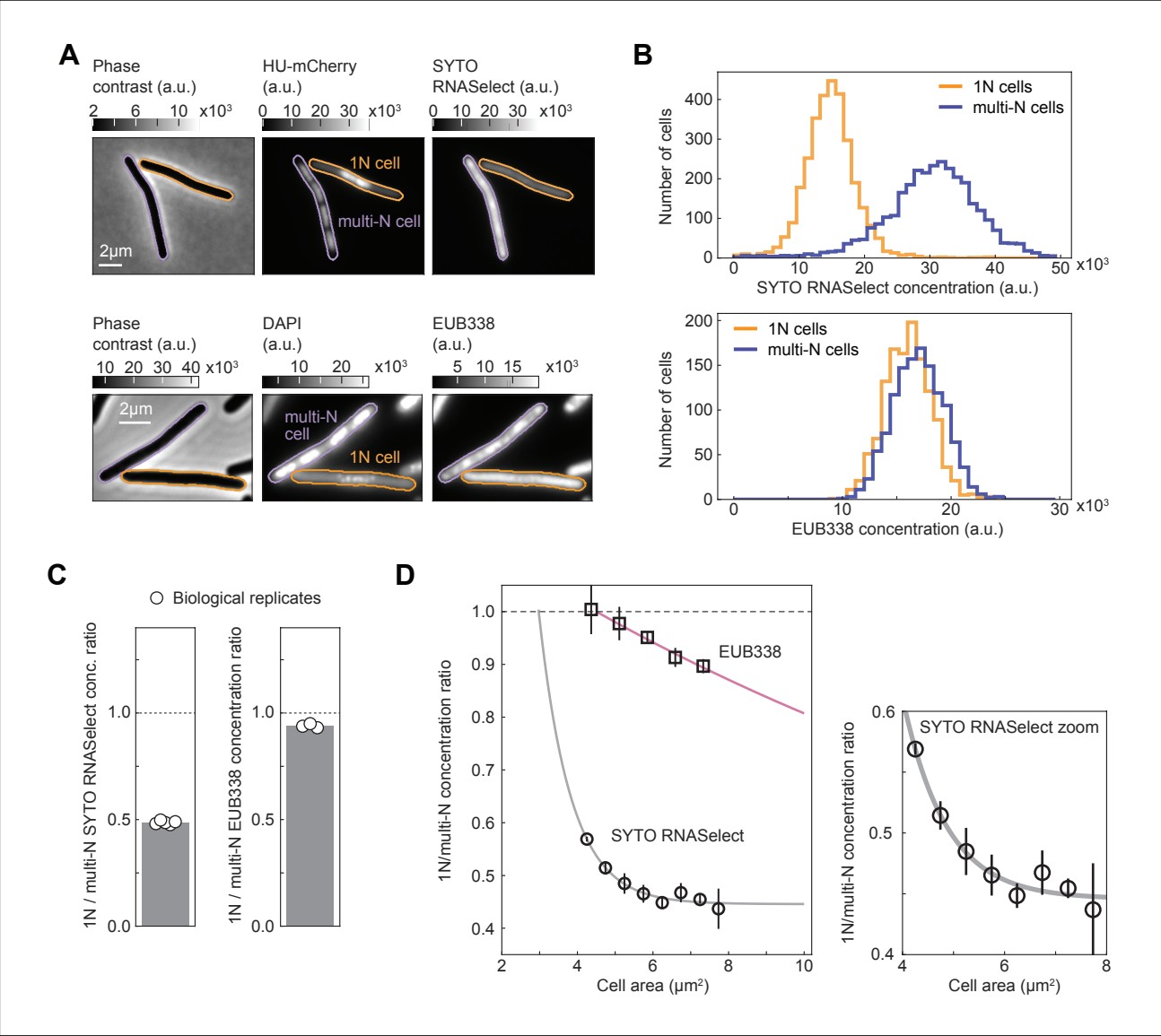

**Figure 4.** RNASelect and EUB338 concentration measurements in 1N and multi-N cells. (**A**) Images of representative cells from a mixed population of 1N (CRISPR interference [CRISPRi] *oriC*) and multi-N (CRISPRi *ftsZ*) cells. Strains CJW7457 and CJW7576 carrying HU-mCherry were used for the SYTO RNASelect staining experiment, whereas DAPI-stained strains SJ_XTL676 and SJ_XTL229 were used for the EUB338 ribosomal RNA (rRNA) fluorescence in situ hybridization (FISH) experiment. (**B**) Concentration distribution of SYTO RNASelect (3077 cells for each population from five biological replicates) and EUB338 (1254 cells for each population from three biological replicates) in 1N and multi-N cells. (**C**) The average 1N/multi-N SYTO RNASelect and EUB338 concentration ratio (gray bar) calculated from five and three biological replicates (white circles), respectively. (**D**) RNASelect and EUB338 concentration ratios as functions of cell area (mean ± SD from five and three biological replicates, respectively). Single exponential decay functions were fitted to the average ratios (*R²*>97%) for each indicated reporter. All concentration comparisons or ratio calculations were performed for equal numbers of 1N and multi-N cells and overlapping cell area distributions (see Materials and methods and *Figure 4—figure supplement 1*).

The online version of this article includes the following figure supplement(s) for figure 4:

**Figure supplement 1.** Cell sampling to match cell size distribution in mixed populations of 1N and multi-N cells.

**Figure supplement 2.** Comparison of RpoC-HaloTag-JF549 labeling between 1N and multi-N cells.

**Figure supplement 3.** EUB338 staining comparison between fast (M9glyCAAT) and slow (M9gly) growing populations.

## Genome dilution rapidly limits RNAP activity under both nutrient-rich and -poor conditions, but the extent of downstream effects on ribosome activity and cell growth can vary with the nutrient condition

In the relatively nutrient-rich M9glyCAAT condition, WT cells at birth are expected to have higher

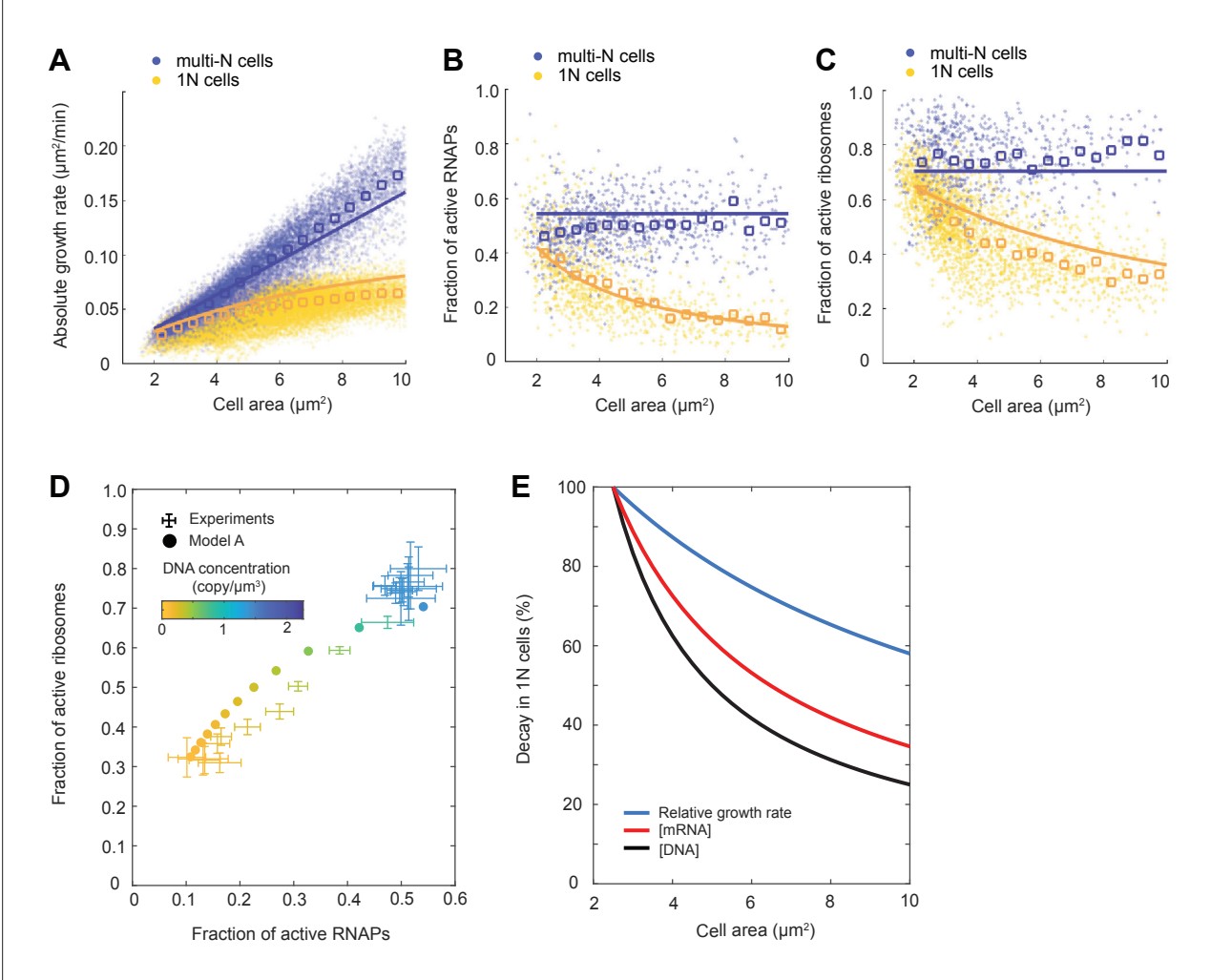

**Figure 5.** Mathematical modeling of DNA limitation. (**A–C**) Plots comparing simulation results of model A (solid lines) with experimental data points (dots) and averages (open squares) in the M9glyCAAT condition. The multi-N and 1N cells are indicated as blue and yellow, respectively: (**A**) The relation between the absolute growth rate ($\frac{dA}{dt}$) and cell area ($A$). (**B**) The relation between the active RNA polymerase (RNAP) fraction and cell area. (**C**) The relation between the active ribosome fraction and cell area. (**D**) Diagram showing how the fractions of active RNAPs and ribosomes change with DNA concentration (colored from yellow to blue). Simulated results (filled dots) are based on model A. Experimental data (points with 2D error bars: 95% CI) from multi-N and 1N cells were combined and shown in the same plot. (**E**) Plot showing the effect of DNA limitation (using the ordinary differential equation [ODE] model A) on the decay of DNA concentration, mRNA concentration, and relative growth rate in 1N cells. Each quantity was normalized to their value at normal cell size (cell area = 2.5 μm²).

The online version of this article includes the following figure supplement(s) for figure 5:

**Figure supplement 1.** Comparison between experimental results from the M9glyCAAT condition and simulation results using model B.

**Figure supplement 2.** Model A-based simulations examining the effects of varying the rates in either mRNA synthesis or mRNA degradation on the relative growth rate of 1N cells as a function of cell area.

DNA content than 1N cells on average due to overlapping DNA replication (*Fossum et al., 2007*). To examine whether cells are also subject to DNA-limited transcription when multi-fork DNA replication is rare or nonexistent, we examined the total RNAP activity of 1N cells relative to WT cells in two different nutrient-poor media, M9gly and M9 L-alanine (M9ala). Abundance and diffusivity measurements of RpoC-labeled RNAPs (*Figure 6—figure supplement 1*) showed that the scaling between the total amount of active RNAPs (i.e. global transcriptional activity) and cell area was strongly reduced in 1N cells, even within the range of WT cell sizes (*Figure 6A and B*). Thus, genome dilution rapidly limits global transcription in nutrient-poor (slow growth) conditions, as in richer (faster growth) conditions (*Figure 3H*).

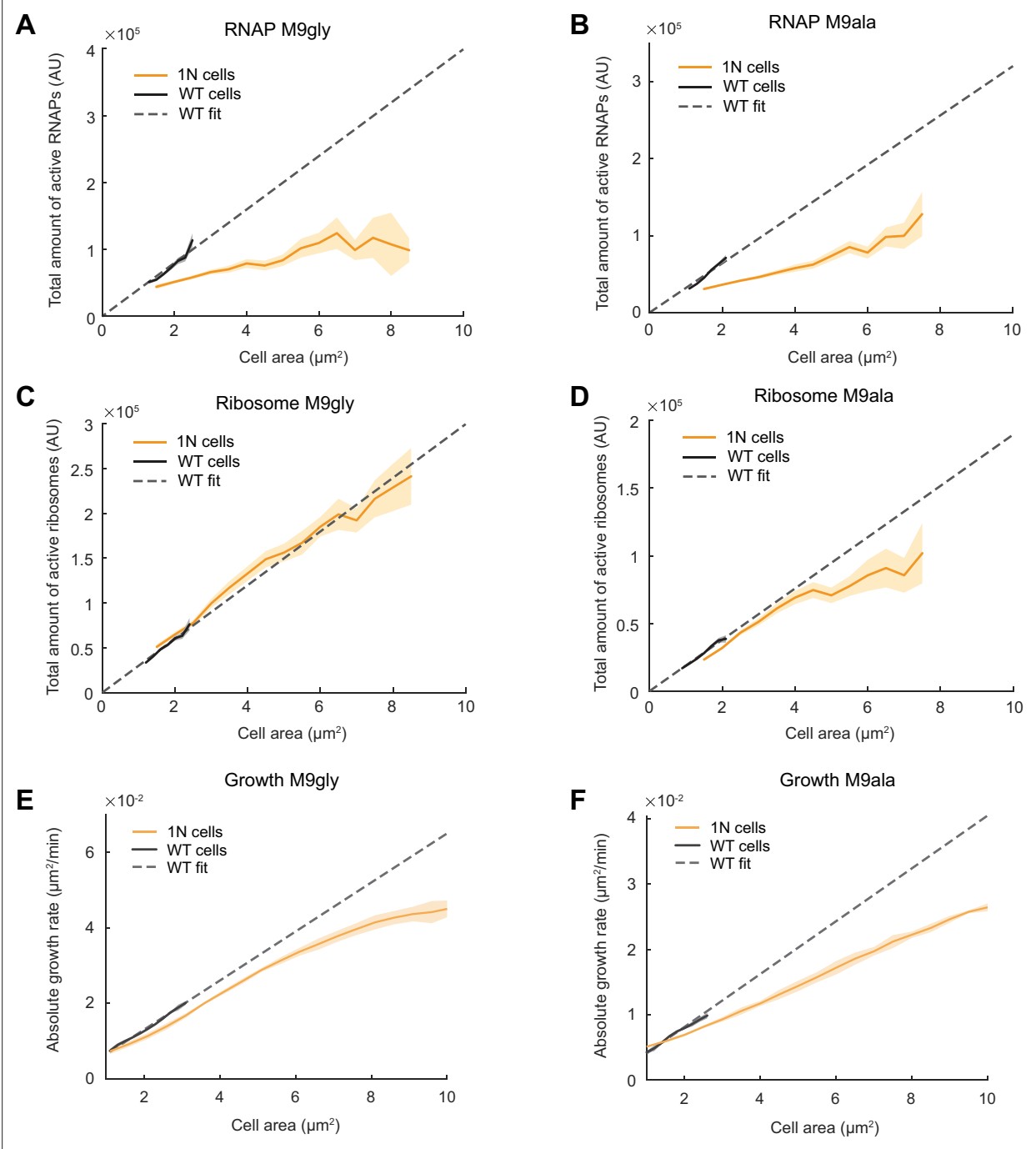

**Figure 6.** Scaling of the total active RNA polymerases (RNAPs), total active ribosomes, and growth rate with cell area during genome dilution in nutrient-poor media. (**A**) Plot showing the total amount of active RNAPs (calculated by multiplying the total amount of RNAPs by the fraction of active RNAPs from **Figure 6—figure supplement 1A and G**) in wild-type (WT) (CJW7339) and 1N (CJW7457) cells grown in M9gly as a function of cell area. Also shown is a linear fit to WT data ($f(x) = 3.99 \cdot 10^4 \cdot x$, $R^2$=0.90). Shaded areas denote 95% CI of the mean from bootstrapping. All data are from three biological replicates. (**B**) Same as (**A**) but for cells grown in M9ala (calculated from **Figure 6—figure supplement 1B and H**). The linear fit for WT data is $f(x) = 3.21 \cdot 10^4 \cdot x$, $R^2$=0.95. (**C**) Plot showing the total active ribosome amount of 1N and multi-N cells grown in M9gly as a function of cell area. The total amount of active ribosomes was calculated by multiplying the total amount of ribosomes by the fraction of active ribosomes (from **Figure 6—figure supplement 2A and G**). Also shown is a linear fit to WT data ($f(x) = 2.99 \cdot 10^4 \cdot x$, $R^2$=0.97). Lines and shaded areas denote mean and 95% CI of the mean from bootstrapping. All data are from three biological replicates. (**D**) Same as (**C**) but for cells grown in M9ala (calculated from **Figure 6—figure supplement 2B and H**). Here, the linear fit to the WT data is $f(x) = 1.90 \cdot 10^4 \cdot x$, $R^2$=0.99. (**E**) Absolute growth rate in 1N (50,352 datapoints from 973 cells) and WT (80,269 datapoints from 12,544 cells) cells in M9gly. The linear fit for WT data is $f(x) = 6.50 \cdot 10^{-3} \cdot x$, $R^2$=0.99.

*Figure 6 continued on next page*

*Figure 6 continued*

(**F**) Absolute growth rate in 1N (71,736 datapoints from 909 cells) and WT (63,367 datapoints from 6880 cells) cells in M9ala. The linear fit for WT data is $f(x) = 4.05 \cdot 10^{-3} \cdot x$, $R^2$=0.97. Lines and shaded areas denote mean ± SD from three biological replicates.

The online version of this article includes the following figure supplement(s) for figure 6:

**Figure supplement 1.** Characterization of RNA polymerase (RNAP) diffusion and active fraction in poor media conditions.

**Figure supplement 2.** Characterization of ribosomal diffusion and active fraction in poor media conditions.

In contrast, abundance and diffusivity measurements of fluorescently labeled ribosomes in cells growing in M9gly and M9ala (*Figure 6—figure supplement 2*) revealed that the total amount of active ribosomes (i.e. bulk translational activity) and the absolute growth rate of 1N cells started to deviate from proportional scaling with cell areas mostly when cells reached large (non-physiological) sizes (*Figure 6C-F*). As a result, the difference in absolute growth rate between 1N and multi-N cells was not as pronounced as in cells growing in the richer M9glyCAAT medium (*Figure 6E and F* vs. *Figure 1C*). This suggests that one or more cellular buffering activities may help mitigate the limitation of DNA concentration on transcription in nutrient-poor media (see Discussion).

## Genome dilution changes the composition of the transcriptome and proteome

The fact that the relative concentrations of ribosomal proteins and RNAP subunits scaled differently with cell area in 1N cells (*Figures 2A, B and 3A, B*) indicated that all genes are not equally impacted by DNA dilution. In yeast and mammalian cells, a decrease in the DNA-per-volume ratio has recently been demonstrated to alter the composition of the proteome, with some proteins increasing in relative concentration while others become comparatively more diluted (*Lanz et al., 2024*; *Lanz et al., 2022*). To examine whether this effect may be conserved across domains of life, we used our proteomic TMT-MS data on the CRISPRi strains to quantify the relative concentration of each detected protein across cell areas following DNA replication or cell division arrest in M9glyCAAT. For each protein, we calculated the relative change in concentration against the relative change in cell size through regression fitting, yielding a slope value. A slope of zero indicates that the concentration of a protein remains constant relative to the proteome whereas a slope of –1 (or 1) means that the relative concentration is decreasing (or increasing) by twofold with each cell size doubling (*Figure 7A*).

We found that the slope distribution was highly reproducible between biological replicates (*Figure 7—figure supplement 1A and B*) but drastically different between 1N-rich cells and multi-N cells (*Figure 7B*, *Supplementary file 1*). In the control multi-N cells where the genome concentration does not change with cell growth, the relative concentration of ~94% of the detected proteins (2217/2360) remained roughly constant, with their relative concentrations decreasing or increasing by less than 20% per cell size doubling (i.e. slopes>–0.2 or <0.2; *Figure 7B* and *Supplementary file 1*). This suggests that protein amounts largely scale with cell size, as generally assumed. However, in 1N-rich cells where the genome dilutes with cell growth, the proportion of detected proteins with slopes near zero (>–0.2 or <0.2) dropped to ~37% (859/2360) (*Figure 7B* and *Supplementary file 1*). A principal component analysis on the relative protein concentration during cell growth confirmed that the relative proteome composition changed proportionally with genome dilution (1N-rich cells), whereas it remained constant when the DNA-to-cell volume ratio was maintained (multi-N cells) (*Figure 7C*).

To examine whether the proteome scaling behavior stems from differential changes in mRNA levels, we performed transcriptomic (RNA-seq) analysis on two biological replicates of 1N-rich cells at different time points after induction of DNA replication arrest. The two replicates were strongly correlated at the transcript level (Spearman $\rho$=0.91, p-value<$10^{-10}$, *Figure 7—figure supplement 1C*). We also found a strong correlation (Spearman $\rho$=0.76, p-value<$10^{-10}$) in scaling behavior with cell area between mRNAs and proteins across the genome of 1N-rich cells (*Figure 7D* and *Supplementary file 4*), indicating that most of the changes in protein levels observed upon genome dilution take place at the mRNA level.

To investigate whether central processes may contribute to the observed transcriptome remodeling during DNA limitation, we examined whether the RNA slopes correlate with gene-specific rates of transcription initiation or mRNA degradation obtained from a published dataset (*Balakrishnan*

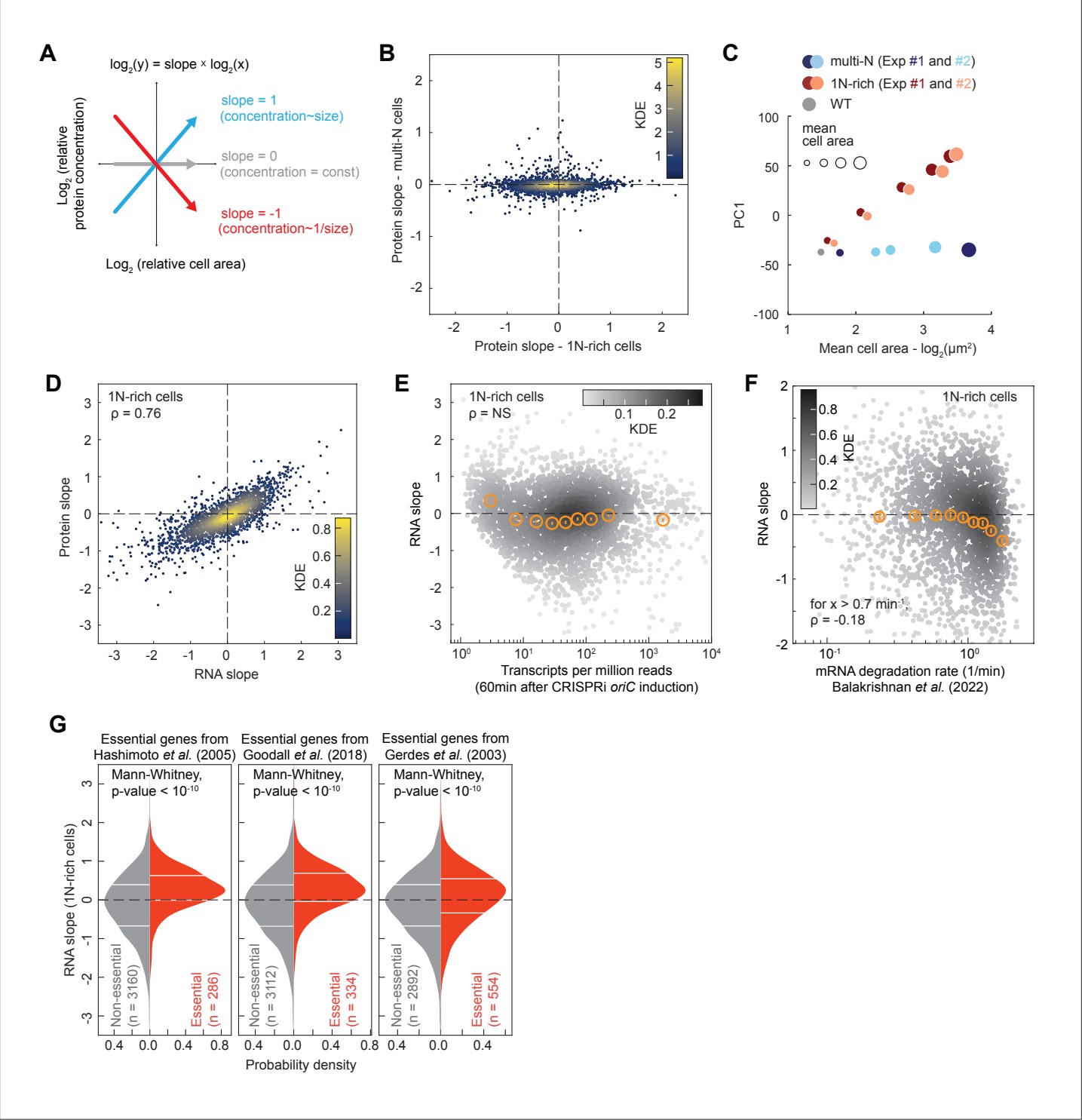

**Figure 7.** Proteome and transcriptome remodeling in 1N-rich cells. (**A**) Schematic explaining the calculation of the protein slopes, which describes the scaling of the relative protein concentration (concentration of a given protein relative to the proteome) with cell area. (**B**) Plot showing the protein scaling (average slopes from two reproducible biological replicates, see *Figure 7—figure supplement 1A and B*) in 1N (x-axis) and multi-N (y-axis) cells across the detected proteome (2360 proteins). The colormap corresponds to a Gaussian kernel density estimation (KDE). (**C**) Plot showing the first principal component (PC1) used to reduce the dimensionality of the relative protein concentration during cell growth. The PC1, which represents the overall change in relative concentration regardless of the sign of the slope, explains 69% of the total variance considering both 1N-rich and multi-N cells. The x-axis corresponds to the log-transformed cell area, whereas the marker size shows the cell area increase in linear scale. (**D**) Correlation between average protein and RNA slopes across 2324 genes. The colormap corresponds to a KDE. (**E**) Relation between mRNA abundance (transcripts

*Figure 7 continued on next page*

*Figure 7 continued*

per million 60 min after CRISPR interference [CRISPRi] induction) and RNA slopes in 1N-rich cells. The colormap indicates a KDE (3446 genes in total). The binned data are also shown (orange markers: mean ± standard error of the mean [SEM], ~380 genes per bin). The Spearman correlation ($\rho$=–0.04) is considered not significant (NS, p-value>$10^{-10}$). (**F**) Correlation between RNA slopes and mRNA degradation rate from a published dataset (***Balakrishnan et al., 2022***) across genes. The colormap indicates a KDE (2570 genes with quantified slopes and positive mRNA degradation rates). The binned data are also shown (orange markers: mean ± SEM, ~280 genes per bin). A significant negative Spearman correlation (p-value<$10^{-10}$) is shown for mRNAs with a degradation rate above 0.7 min$^{-1}$. (**G**) RNA slope comparison between essential and non-essential genes in *E. coli*. Three different published sets of essential genes were used (***Gerdes et al., 2003***; ***Goodall et al., 2018***; ***Hashimoto et al., 2005***). The horizontal white lines indicate the inter-quartile range of each distribution. Mann-Whitney non-parametric tests justify the significant difference (p-value<$10^{-10}$) between the two gene groups (essential vs. non-essential genes).

The online version of this article includes the following figure supplement(s) for figure 7:

**Figure supplement 1.** Comparison of protein and mRNA scaling between biological replicates.

**Figure supplement 2.** Comparison of our data with reference datasets.

**Figure supplement 3.** Protein slopes relative to the chromosome position of their gene in 1N-rich and multi-N cells.

**Figure supplement 4.** Protein slopes relative to protein ion intensity for 1N-rich cells.

*et al., 2022*). Note that the reference dataset was generated from experiments on *E. coli* growing in M9 glucose (M9glu) and not M9glyCAAT. However, both media give similar growth rates (***Govers et al., 2024***) and our transcriptome measurements agree well with the reference data in terms of mRNA abundance (Spearman $\rho$=0.76, p-value<$10^{-10}$, ***Figure 7—figure supplement 2A***). We found no significant correlation between the rates of transcription initiation from the reference dataset and RNA or protein slopes across genes (***Figure 7—figure supplement 2B and C***). Consistent with this finding, mRNA abundance was not a predictor of RNA slopes (***Figure 7E***). This was somewhat surprising as one might anticipate highly transcribed genes to saturate with RNAPs faster than other genes. However, we found that the mRNA degradation rate partly explains the variance in RNA and protein slopes. Specifically, for genes producing short-lived transcripts (decay rate>0.7 min$^{-1}$), the RNA and protein slopes slightly negatively correlated with the rate of mRNA decay (Spearman correlation coefficient $\rho$=–0.18, p-value<$10^{-10}$, ***Figure 7F*** and ***Figure 7—figure supplement 2D***, ***Supplementary file 5***). These results suggest that genes that generate short-lived mRNAs are more susceptible to DNA limitation, presumably because their mRNAs are more rapidly diluted with cell growth due to their fast decay, though we cannot exclude potential indirect effects.

Next, we examined whether genes reported to be essential for viability in three independent studies (***Gerdes et al., 2003***; ***Goodall et al., 2018***; ***Yamazaki et al., 2008***) displayed biases in RNA and protein slopes given the importance of their products for cell growth. Remarkably, essential genes, which share similar mRNA decay rates as other genes (***Figure 7—figure supplement 2E***, Mann-Whittney p-value>0.01), tended to exhibit superscaling behavior in 1N cells as shown by their enrichment in positive RNA slopes regardless of the selected dataset (***Figure 7G***, Mann-Whittney p-value<$10^{-10}$). This suggests that cells have evolved regulatory mechanisms to minimize dilution of mRNAs encoded by essential genes.

## Discussion

Our data suggest that DNA limitation in *E. coli* cells affects cell growth rate through modulation of downstream transcription and translation activities (***Figures 1–7*** and associated figure supplements). The fact that DNA limitation for cellular growth was also observed in *C. crescentus* (***Figure 1—figure supplement 8***) is significant not only because this bacterium is distantly related to *E. coli*, but also because it has a different pattern of cell wall growth and distinct control mechanisms of DNA replication (***Aaron et al., 2007***; ***Banerjee et al., 2017***; ***Frandi and Collier, 2019***; ***Lasker et al., 2016***; ***Terrana and Newton, 1975***). This suggests that DNA concentration may be a prevalent growth constraint across bacterial species. It also helps explain why the timing of DNA replication in bacteria is so robustly linked to cell volume across environmental and genetic conditions that affect cell size (***Donachie, 1968***; ***Govers et al., 2024***; ***Sauls et al., 2019***; ***Si et al., 2017***; ***Zheng et al., 2016***).

Comparison with studies on eukaryotic cells suggests conservation of gene expression principles across domains of life. For instance, in yeast, it has been shown that the global transcription rate in G1-arrested cells is higher in diploids than haploids of similar sizes (***Swaffer et al., 2023***),

consistent with DNA concentration being a limiting factor for transcription. Furthermore, in both yeast and mammalian cells, small G1-arrested cells display higher growth rate (or global RNA or protein synthesis rate) per cell volume than large ones that have exceeded a certain volume (*Cadart et al., 2018*; *Liu et al., 2024*; *Neurohr et al., 2019*; *Lanz et al., 2022*). This is likely due to a change in genome concentration rather than a change in cell volume, as the relative growth rate is unaffected in very large cells as long as they undergo a proportional increase in ploidy (*Virtanen et al., 2020*).

We found that even a relatively small dilution in DNA concentration—as expected in DNA replication-arrested *E. coli* cells that are still within or close to physiological sizes—results in a reduction of total RNAP activity in both rich and poor media (*Figures 3H and 6A, B*). Crude estimations suggest that ≤40% DNA dilution is sufficient to negatively affect transcription (total RNAP activity) in M9glyCAAT, whereas the same effect was observed after less than ~10% dilution in poor media (M9gly or M9ala) (see Materials and methods). Thus, cells appear to live at the cusp of DNA limitation for transcription, especially under slow growth (nutrient-poor) conditions. This suggests that cells make enough—but not too much—DNA, presumably because DNA replication is a costly process that represents a significant fraction (~6% in minimal media) of the cellular energy budget (*Neidhardt et al., 1990*).

What may be the implications of living close to DNA limitation? While *E. coli* carefully controls its genome concentration across various conditions and growth rates at the population level (*Donachie, 1968*; *Govers et al., 2024*; *Si et al., 2017*; *Zheng et al., 2016*), there remains variability in DNA concentration at the single-cell level, with some cells initiating DNA replication at smaller or larger cell volumes than others (*Si et al., 2019*; *Witz et al., 2019*). In future studies, it will be interesting to explore whether this variability contributes to the known growth rate heterogeneity across isogenic cells (*Lin and Jacobs-Wagner, 2022*; *Wang et al., 2010*). It is also tempting to speculate that changes in genome concentration may, at least in part, contribute to the deviations from exponential growth that have been reported during the division cycle of *B. subtilis*, *E. coli,* and stalked *C. crescentus* progeny (*Banerjee et al., 2017*; *Kar et al., 2021*; *Nordholt et al., 2020*; *Reshes et al., 2008*). More substantial forms of DNA dilution may occur under other circumstances. *C. crescentus* cells in freshwater lakes often form long filaments during algal blooms in the summer months (*Heinrich et al., 2019*). These filament cells are thought to be the result of a DNA replication arrest in response to the combination of an alkaline pH, a depletion in phosphate, and an excess of ammonium (*Heinrich et al., 2019*). Another example is illustrated by the Lyme disease agent *Borrelia burgdorferi*. This pathogen, which forms long polyploid cells during exponential growth, experiences a progressive decrease in genome concentration (up to eightfold) in stationary phase laboratory cultures through the gradual loss of genome copies (*Takacs et al., 2022*).

In yeast cells, decreased mRNA turnover combined with increased RNAP II gene occupancy helps mitigate DNA dilution on global transcriptional activities up to a certain (non-physiological) cell volume, beyond which the compensation breaks down (*Swaffer et al., 2023*; *Zhurinsky et al., 2010*). Such buffering activities, which are consistent with model predictions (*Figure 5—figure supplement 2*; *Swaffer et al., 2023*), may also be at play in *E. coli* in a growth medium-dependent manner. While genome dilution rapidly impacted transcription in all tested media based on total RNAP activity measurements (*Figures 3H and 6A, B*), we found that the negative impact on downstream processes—total ribosome activity and cell growth—occurred later (i.e. mostly beyond physiological cell sizes) in M9gly and M9ala (*Figure 6C–F*), in contrast to M9glyCAAT (*Figures 1B, C and 2H*). This suggests the existence of mechanisms that compensate for DNA-limited transcription under slow growth such as a decrease in mRNA decay, an increase in ribosome loading, and/or an increase in translation elongation rate. Perhaps such buffering activities are not as effective under nutrient-rich conditions due to the rapid mRNA dilution during fast growth. Testing these hypotheses will require future experimentation.

Another remarkable similarity between bacteria and eukaryotes is the effect of genome concentration on proteome composition. While protein abundance is typically assumed to scale with cell size in bacteria, we found that this is true at the proteome level only when ploidy also scales (*Figure 7B*). This requirement was also recently shown in yeast and mammalian cells (*Lanz et al., 2024*; *Lanz et al., 2022*). This conservation of scaling principle further highlights the importance of genome concentration in controlling protein expression.

What determines the scaling behavior of proteins in *E. coli* is not clear. We found that it largely occurs at the mRNA level (*Figure 7D*), and that short-lived mRNAs are slightly more susceptible to subscaling behavior (*Figure 7F* and *Figure 7—figure supplement 2D*). Conversely, the majority of essential genes (*Gerdes et al., 2003*; *Goodall et al., 2018*; *Yamazaki et al., 2008*) tended to display superscaling behavior relative to the rest of the genome (*Figure 7G*, *Supplementary file 5*). This suggests the existence of regulatory mechanisms that prioritize the expression of essential genes over less important ones when genome concentration becomes limiting for cell growth.

While the scaling of proteins in 1N cells is largely driven by that of mRNAs (*Figure 7D*), we found that protein slopes, but not RNA slopes, displayed a slight yet significant positive correlation (Spearman $\rho$=0.23, p-value<$10^{-10}$) with *oriC* proximity for genes within 1.35 Mb from *oriC* (*Figure 7—figure supplement 3A and B*). Why and how this occurs is unclear, but it suggests that mRNA-independent mechanisms (i.e. independent of mRNA synthesis or decay) also contribute to protein scaling behavior. At the GO term level, we did not identify any specific trends in proteome changes (*Supplementary file 1*). In eukaryotic cells, histones are known to scale in proportion with DNA rather than cell size (*Claude et al., 2021*; *Swaffer et al., 2023*; *Wiśniewski et al., 2014*). As a result, their concentration proportionally decreases (i.e. slope = –1) with growth in G1 phase. In *E. coli*, the relative abundance of some nucleoid-associated proteins (H-NS, HU, and Dps) decreased with genome dilution, while others (IHF and Fis) displayed superscaling (protein slopes > 0) behavior (*Figure 7—figure supplement 4*).

Given the prevalent use of *E. coli* in the biotechnological world, we hope that our findings will be helpful to future bioengineering studies and growth rate optimization efforts. We show that protein content and cellular growth depend on the ploidy-to-cell volume ratio (*Figures 1 and 7*). As such, models of protein expression that take into consideration the DNA concentration and the active number of RNAPs and ribosomes could provide a starting point to identify the parameter space that leads to growth rate improvement. Experimentally, it will be important to determine which specific genes exert the largest growth rate-limiting effect. In this context, the few essential genes with strong subscaling behavior (large negative values of RNA and protein slopes) in 1N cells (*Figure 7G*, *Supplementary file 5*) suggest potential candidates for future studies given the rapid dilution of their mRNAs and proteins relative to other genes.

## Materials and methods
### Bacterial strains and growth conditions
Bacterial strains are listed in *Supplementary file 6*, which includes their sources (*Bakshi et al., 2012*; *Gray et al., 2019*; *Guyer et al., 1981*; *Nielsen et al., 2006*; *Takacs et al., 2018*; *Thanbichler and Shapiro, 2006*; *Thanbichler et al., 2007*; *Xiao et al., 1991*; *West et al., 2002*) and methods of construction (*Datsenko and Wanner, 2000*). Oligomers used for polymerase chain reaction are listed in *Supplementary file 7*. Transductions and Gibson assemblies were performed as described previously (*Ely, 1991*; *Thomason et al., 2007*).

*E. coli* strains were grown at 37°C in M9 minimal media with different supplements: 0.2% glycerol, 0.1% casamino acids, and 1 μg/mL thiamine (M9glyCAAT), 0.2% glycerol (M9gly), or 0.2% L-alanine (M9ala). For microscopy, cells were grown in culture tubes to stationary phase, diluted 10,000-fold, and grown until they reached an optical density (OD$_{600}$) between 0.05 and 0.2. For imaging, cells were then spotted onto a 1% agarose pad on a glass slide prepared with the appropriate M9 medium and covered by a #1.5 thickness coverslip.

For time-lapse microscopy experiments with the CRISPRi strains (*Li et al., 2016*), the cells were induced by adding L-arabinose (0.2%) to liquid cultures after which a sample (~1 μL) was immediately collected and spotted on an agarose pad containing the appropriate growth medium supplemented with 0.2% L-arabinose. This was promptly followed by imaging of individual cells. To determine the 1N status of CRISPRi *oriC* cells, we monitored the last division and number of nucleoids based on HU-m-Cherry fluorescence. In *Figure 1B*, the time point '0 min' refers to the time when cells have reached the 1N status.

For population microscopy experiments with the CRISPRi strains, the cells were induced with 0.2% L-arabinose and allowed to grow in normal conditions until a specific time point (depending on the growth medium and strain) was reached, after which the cells were spotted on an agarose pad containing the appropriate growth medium and 0.2% L-arabinose, and imaged (see information for

each experiment in the corresponding figure). Note that the CRISPRi strains do not metabolize arabinose due to the *araBAD* deletion.

For the TMT-MS experiments, CRISPRi *oriC* (SJ_XTL676) cells were supplemented with 0.2% L-arabinose and allowed to grow in liquid M9glyCAAT cultures at 37°C for 0, 120, 180, 240, and 300 min before harvesting, while CRISPRi *ftsZ* (SJ_XTL229) cells were collected after 0, 60, and 120 min after arabinose addition. For RNA-seq experiments, CRISPRi *oriC* (SJ_XTL676) cells were supplemented with 0.2% L-arabinose and allowed to grow in liquid M9glyCAAT cultures at 37°C for 60, 120 200, and 240 min before harvesting.

Strains carrying the *dnaC2* mutation were grown at a permissive temperature of 30°C and then shifted to 37°C to block replication initiation. Transcription inhibition and mRNA depletion were achieved by exposing cells to 200 µg/mL rifampicin for 30 min before spotting cells on a 1% agarose pad containing the appropriate M9 medium and rifampicin concentration. The HaloTag was labeled with Janelia Fluor 549 (JF549) ligand (*Grimm et al., 2015*) as described previously (*Banaz et al., 2019*). Briefly, cells were incubated with 2.5 µM of the JF549 ligand for 30 min while shaking, washed five times with growth medium, and allowed to recover for several generations (while remaining in exponential phase) prior to imaging.

*C. crescentus* strains were grown at 22°C in PYE (2 g/L bacto-peptone, 1 g/L yeast extract, 1 mM MgSO$_4$, 0.5 mM CaCl$_2$) supplemented with 0.03% xylose to induce production of the essential protein FtsZ (*Wang et al., 2001*) or DnaA (*Gorbatyuk and Marczynski, 2001*). For microscopy, cells were grown in overnight cultures, then diluted at least 1:10,000 in fresh medium and grown to exponential phase (OD$_{660}$ nm<0.3). To induce cell filamentation by depleting FtsZ or DnaA, xylose was removed from the medium by pelleting cells, washing twice with PYE, and resuspending in fresh PYE. Cultures were then allowed to grow for an additional 30 min to allow ongoing cell division cycles to complete before spotting on 1% agarose pads containing PYE but lacking xylose to deplete FtsZ or DnaA. To estimate the concentration of fluorescently labeled ribosomes and RNAPs by fluorescence microscopy, cultures were sampled at 0, 4, 8, and sometimes 12 hr following xylose depletion with 30 min outgrowth to allow late predivisional cells time to divide before spotting. For time-lapse microscopy, new pads were spotted with cells from the original culture at 0, 4, 8, and sometimes 12 hr following xylose removal with 30 min outgrowth in a liquid PYE medium.

## Epifluorescence microscopy

For *E. coli*, phase contrast and fluorescence imaging (except for the RpoC-HaloTag-JF549 epifluorescence experiment) were performed on a Nikon Ti2 microscope equipped with a Perfect Focus System, a 100× Plan Apo $\lambda$ 1.45 NA oil immersion objective, a motorized stage, a Prime BSI sCMOS camera (Photometrics), and a temperature chamber (Okolabs). Fluorescence emission was collected during a 100 or 200 ms exposure time provided by a Spectra III Light Engine LED excitation source (Lumencor): mCherry—594 nm excitation, DAPI/FITC/TxRed filter cube (polychroic FF-409/493/596-Di02, triple-pass emitter FF-1-432/523/702-25), GFP and SYTO RNASelect—488 nm excitation, DAPI/FITC/TxRed polychroic filter cube, and an ET525/50M emission filter; YFP—514 nm excitation, CFP/YFP/mCherry filter cube (polychroic FF-459/526/596-Di01, triple-pass emitter FF-1-475/543/702-25), and a FF02-525/40-25 emission filter. The microscope was controlled using NIS-Elements AR. For time-lapse imaging, phase images were collected every 5 min.

Epifluorescence snapshots of RpoC-HaloTag-JF549 were taken using a Nikon Ti microscope, equipped with a Perfect Focus System, a 100× Plan Apo $\lambda$ 1.45 NA oil immersion objective, a motorized stage, and an ORCA Flash 4.0 camera (Hamamatsu). Fluorescence emission was collected during a 200 ms exposure time provided by a Sola solid-state white light source (Lumencor) and a Cy3 filter cube (excitation AT545/25×, dichroic T565lpxr, emission ET605/70m). The microscope was controlled using NIS-Elements AR. The same microscope and filters were used to capture the EUB338-Cy3 fluorescence, but using a 100 ms exposure time. The same microscope, with a DAPI filter cube (excitation ET395/25×, dichroic T425lpxr, emission ET460/50 m) and 500 ms exposure time, was used to capture the DAPI fluorescence in fixed 1N and multi-N cells.

For *C. crescentus*, phase contrast and fluorescence imaging were performed on a Nikon Ti-E microscope equipped with a Perfect Focus System, a 100× Plan Apo $\lambda$ 1.45 NA oil immersion objective, a motorized stage, an Orca-Flash4.0 V2 142 CMOS camera (Hamamatsu) at room temperature. Chroma filter sets were used to acquire fluorescence images: CFP (excitation ET436/20×, dichroic T455lp,

emission ET480/40m) and mCherry (excitation ET560/40×, dichroic T585lpxr, emission ET630/75m). The microscope was controlled using NIS-Elements AR. For time-lapse imaging, phase images were collected every 2.5 min.

## Photoactivated localization microscopy

For single-molecule photoactivated localization microscopy (PALM), coverslips were plasma-cleaned of background fluorescent particles using a plasma cleaner (PDC-32G, Harrick Plasma). Live-cell PALM was performed on a Nikon N-STORM microscope equipped with a Perfect Focus System and a motorized stage. JF549 fluorescence was measured using an iXon3 DU897 EMCCD camera (Andor) and excited from a 50 mW 561 nm laser (MLC400B laser unit, Agilent) with 50% transmission. The laser was focused through a 100× Apo TIRF 1.49 NA oil immersion objective (Nikon) onto the sample using an angle for highly inclined thin illumination to reduce background fluorescence (*Tokunaga et al., 2008*). Fluorescence emission was filtered by a C-N Storm 405/488/561/647 laser quad set. Transmission illumination was used to gather bright-field images. PALM movies of 20,000 frames were acquired with continuous laser illumination and a camera frame time of 10.7 ms.

## SYTO RNASelect staining experiments

In order to compare the mRNA concentration between 1N and multi-N cells, exponentially growing CJW7576 (CRISPRi *ftsZ*) and CJW7457 (CRISPRi *oriC*) cells were stained with the fluorogenic SYTO RNASelect dye (Invitrogen, S7576) after CRISPRi induction with 0.2% L-arabinose. To ensure overlapping cell area distributions in the absence of cell division, considering the measured growth rate differences between the 1N and multi-N cells, CRISPRi *oriC* was induced for 3.5–4 hr whereas CRISPRi *ftsZ* was induced for 1.5–2 hr. Then, the two populations were mixed at equal optical densities (OD$_{600}$) and stained with 0.5 µM SYTO RNASelect for 15 min at 37°C with shaking. For each staining, a fresh 5 µM SYTO RNASelect stock was prepared in L-arabinose-containing medium. Stained cells (~0.5 µL) were spotted on a 1% agarose pad prepared with the same growth medium with L-arabinose for imaging. Five biological replicates were performed.

## RpoC-HaloTag-JF549 staining experiments for epifluorescence snapshots

For *Figure 4—figure supplement 2*, CRISPRi *oriC* in CJW7520 cells was induced for 3.5 hr whereas CRISPRi *ftsZ* (CJW7527) was induced for 1.5 hr with 0.2% L-arabinose to obtain a similar range of cell sizes for imaging. Then, the two populations were mixed at equal optical densities (OD$_{600}$) and stained with 2.5 µM JF549 at 37°C for 30 min with shaking. The mixed cells were then washed three times with L-arabinose-containing (0.2%) medium. All washes were performed at 4°C, using ice-cold medium to block cell growth and avoid dilution of the dye. Then, the cells (~0.5 µL) were spotted on a 1% agarose pad prepared with the same L-arabinose-containing growth medium for imaging. Two biological replicates were performed.

## FISH experiments

FISH with the EUB338 DNA probe (*Amann et al., 1990*) was used to compare the concentration of 16S rRNAs in fixed *E. coli* 1N (SJ_XTL676) vs. multi-N (SJ_XTL229) cells, or between fast (M9glyCAAT) and slow (M9gly) growing WT MG1655 populations. Similar to the SYTO RNASelect experiments, CRISPRi *oriC* was induced for 3.5–4 hr whereas CRISPRi *ftsZ* was induced for 1.5–2 hr to ensure overlapping cell area distributions between 1N and multi-N cells considering the measured growth rate differences between the two strains. Then, the two populations were mixed at equal optical densities (OD$_{600}$) and fixed prior to staining using previously described protocols (*Kim et al., 2019*; *Kim and Jacobs-Wagner, 2018*; *Kim and Vaidya, 2020*). Note that WT cells growing exponentially in different media (M9gly or M9glyCAAT) were fixed and stained separately (separate tubes and coverslips but using the exact same washing, pre-hybridization, and hybridization buffers).

Pre-hybridization was performed for 2 hr at 37°C using a solution that contained 40% formamide, 2× saline-sodium citrate (SSC), 1× vanadyl ribonucleoside complex (VRC), and 0.4% (wt/vol) bovine serum albumin (BSA). Staining was performed for 13 hr at 37°C using a solution that contained 500 nM EUB338-Cy3 Eubacterial probe (Millipore Sigma, MBD0033), 40% formamide, 2× SSC, 1× VRC, 0.4% (wt/vol) BSA, 0.4 mg/mL *E. coli* tRNA, and 10% dextran sulfate. The high probe concentration allowed

for the saturation of the 16S rRNA as previously shown (*Hoshino et al., 2008*), which made it possible to compare rRNA concentrations between different strains and media. After staining, the fixed cells were washed five times with a wash solution (50% formamide, 2× SSC) and ten times with 1× phosphate-buffered saline (PBS). Right before mounting the coverslip on the glass slide for imaging, the 1N/multi-N cells were further stained with 1 μg/mL DAPI.

## Image processing and data analysis

Data analysis was done in MATLAB (Mathworks) (MAIN_pop_analysis.m script for epifluorescence snapshots, MAIN_timelapse.m script for time-lapse movies, and MAIN_mol_tracking.m script for single-molecule tracking experiments), except for segmentation of phase contrast images, which was done in Python (segmentationRun.py) using a convolutional neural network, the Nested-Unet (*Wiktor et al., 2021*; *Zhou et al., 2020*), and the analysis of the SYTO RNASelect and RpoC-HaloTag-JF549 epifluorescence (snapshots_analysis_UNET_version_ND2.py class) that was also performed in Python 3.9 (DNA_limitation_python_environment.yml). The Nested-Unet network was trained for our microscopy setup using PyTorch 1.7.0 and NumPY 1.19.2 (trainerWrapper.py) (*Harris et al., 2020*; *Paszke et al., 2019*).

Cell area masks from segmentation were linked between time-lapse frames based on maximum overlap (trackCells.m). Two masks linked to the same cell area were considered a cell division event. To estimate the absolute growth rate, the cell area over time was smoothed by a sliding-average window of five datapoints (time interval 5 min) and the difference in the cell area between consecutive frames was calculated. The relative growth rate was calculated by dividing the absolute growth rate by the cell area. Cell areas were converted into volumes using Oufti's scripts (*Paintdakhi et al., 2016*). To avoid bias from cells reaching sizes too large to support growth, the time-lapse data for filamenting cells was truncated based on their maximum absolute growth rate.

Fluorescent ParB-mCherry spots were detected by fitting a 2D Gaussian function to raw image data (detectSpots.m). First, the fluorescent image was filtered using a bandpass filter to identify the local maxima. Next, the local maxima were fitted by a Gaussian function and a spot quality score was calculated based on spot intensity and quality of a Gaussian fit ($Intensity \cdot quality_{fit}/sigma_{fit}$). The spot score threshold was determined by visual inspection of the training data and was set to remove poor-quality spots from analysis.

The number of HU-mCherry-labeled nucleoid areas was determined using Otsu's thresholding (multithresh.m) (*Otsu, 1979*). Minimum and maximum area thresholds for an individual nucleoid were determined by measuring the number of fluorescent spots of ParB protein fusion in HU-labeled nucleoid areas of a strain (CJW7517) carrying a *parS* site from plasmid pMT1 at *ori1* (*Figure 1— figure supplement 3A*). Only cells containing a single nucleoid were considered 1N cells. To measure the total fluorescence intensity of a cell, the median intensity of the area outside the cell areas was subtracted from the fluorescence intensity of each pixel of a cell, and the intensity of all pixels was summed together.

For the SYTO RNASelect, RpoC-HaloTag-JF549, and EUB338 epifluorescence snapshot experiments (*Figure 4* and *Figure 4—figure supplements 1–3*), the nucleoid objects were segmented using the *segment_nucleoids* function in the *snapshots_analysis_UNET_version_ND2.py* class. This function combines a Laplacian of Gaussian filter, an adaptive filter, and a hard threshold to detect the nucleoid boundaries and distinguish between 1N and multi-N cells in the mixed populations. The image filters were applied using the scikit-image (*van der Walt et al., 2014*) and NumPy (*Harris et al., 2020*) Python libraries. For the SYTO RNASelect-stained cells, the HU-mCherry fluorescence was used to segment the nucleoid objects. In the EUB338 FISH experiments, the nucleoids were stained with DAPI. For the RpoC-HaloTag-JF549-stained cells, the fluorescence of RpoC-HaloTag-JF549 bound to nucleoids was used to segment the nucleoid objects. The number of segmented nucleoid objects was used to distinguish between 1N (one nucleoid object) and multi-N (two or more nucleoid objects) cells. The classification results were curated manually by visual inspection.

The SYTO RNASelect concentration corresponds to the total fluorescence of the fluorogenic dye within the cell boundaries of the cell mask, divided by the area of the cell mask. Similarly, the RpoC-HaloTag-JF549 and EUB338 concentrations were calculated in arbitrary units.

To ensure that the SYTO RNASelect, the RpoC-HaloTag-JF549, or the EUB338-Cy3 fluorescence was compared for the same distributions of cell areas between the 1N and multi-N cells in the mixed

populations, random sampling was performed in each cell area bin using a sample size equal to the smaller cell number between the 1N and multi-N cells (*Figure 4—figure supplements 1 and 2B*). For example, if in each cell area bin, there were 100 1N cells and 500 multi-N cells, 100 cells were randomly sampled from the multi-N population to match the sample size of the 1N population. Bins with less than 25 (for the EUB338 experiments) or 50 (for the RNASelect or the RpoC-HaloTag-JF549 experiments) cells per population across biological replicates were removed from the analysis. This sampling was performed one time to compare the distributions of the SYTO RNASelect, EUB338, and RpoC-HaloTag-JF549 (*Figure 4B and C* and *Figure 4—figure supplement 2C and D*) concentrations between the 1N and multi-N cells. However, to estimate the average 1N/multi-N SYTO RNASelect, EUB338 or RpoC-HaloTag-JF549 ratio during cell growth (*Figure 4D* and *Figure 4—figure supplement 2E*) multiple samplings (with substitution) were performed for each biological replicate and cell area bin. This allowed us to use all the data while still comparing equal numbers of 1N and multi-N cells per biological replicate and cell area bin. Cell area bins with less than 5 (for the EUB338 experiments) or 10 (for the RNASelect or the RpoC-HaloTag-JF549 experiments) cells per biological replicate were removed from the analysis. Also here, the sample size was set by the smallest population size (1N or multi-N population) and the number of iterations was equal to the size-difference between the populations multiplied by 10.

## Single-molecule tracking analysis

Single-molecule tracking data was analyzed as previously described (*Mäkelä and Sherratt, 2020*). Candidate fluorescent spots were detected using band-pass filtering and an intensity threshold for each frame of the time-lapse sequence. These initial localizations were used as starting positions in phasor spot detection for high-precision localization (*Martens et al., 2018*). Individual molecules were then tracked in each cell area by linking positions to a trajectory if they appeared in consecutive frames within a distance of 0.8 µm. Cell areas were detected from bright-field images using Microbe-Tracker (*Sliusarenko et al., 2011*). In the case of multiple localizations within the tracking radius, these localizations were omitted from the analysis. Tracking allowed for a single frame disappearance of the molecule within a trajectory due to blinking or missed localization. The mobility of each molecule was determined by calculating an apparent diffusion coefficient, $D_a$, from the stepwise mean-squared displacement of the trajectory using:

$$D_a = \frac{1}{4n\Delta t} \sum_{i=1}^{n} \left[ x\left(i\Delta t\right) - x\left(i\Delta t + \Delta t\right) \right]^2 + \left[ y\left(i\Delta t\right) - y\left(i\Delta t + \Delta t\right) \right]^2$$

where $x(t)$ and $y(t)$ indicate the coordinates of the molecule at time $t$, $\Delta t$ is the frame rate of the camera, and $n$ is the total number of the steps in the trajectory. Trajectories with less than nine displacements were omitted due to the higher uncertainty in $D_a$.

The calculated $D_a$ values are expected to reflect different dynamic states of molecules. To determine the fraction of molecules in each state, $\log_{10}$-transformed $D_a$ data (*Figure 2D*) was fitted to a Gaussian mixture model (GMM) using the expectation-maximization algorithm (*Bishop, 2006*). A mixture of three Gaussian distributions with free parameters for mean, SD, and weight of each state were fitted for different conditions (*Figure 2D* and *Figure 6—figure supplement 2E and F*). Additionally, for 1N cells, single-molecule tracking data were binned and fitted as a function of cell area (*Figure 2C* and *Figure 6—figure supplement 2C and D*). To determine the active RNAP or ribosome fraction of a single cell, the GMM was used to determine the state of each molecule from the measured $D_a$, and the fraction of molecules in the slowest ('active') state was calculated. Only cells with at least 50 trajectories were considered in the analysis for more accurate quantification. The total quantity of active molecules was estimated by multiplying the measured total fluorescence intensity with the measured active fraction as a function of the cell area.

## Sample preparation for liquid chromatography coupled to tandem MS

CRISPRi *oriC* (SJ_XTL676) cells were collected 0, 120, 180, 240, and 300 min after addition of 0.2% arabinose, while CRISPRi *ftsZ* (SJ_XTL229) cells were collected after 0, 60, and 120 min of induction, and pelleted. Cell pellets were lysed in 1% SDS at 95°C for 10 min (with vigorous intermittent vortexing and in the presence of 5 mM β-mercaptoethanol as a reducing agent). Cell lysates were cleared by centrifugation at 15,000 × *g* for 30 min at 4°C. The lysates were alkylated with 10 mM

iodoacetamide for 15 min at room temperature, and then precipitated with three volumes of a solution containing 50% acetone and 50% ethanol. Precipitated proteins were re-solubilized in 2 M urea, 50 mM Tris-HCl, pH 8.0, and 150 mM NaCl, and then digested with TPCK-treated trypsin (50:1) overnight at 37°C. Trifluoroacetic acid was added to the digested peptides at a final concentration of 0.2%. Peptides were desalted with a Sep-Pak 50 mg C18 column (Waters). The C18 column was conditioned with five column volumes of 80% acetonitrile and 0.1% acetic acid, and then washed with five column volumes of 0.1% trifluoroacetic acid. After samples were loaded, the column was washed with five column volumes of 0.1% acetic acid followed by elution with four column volumes of 80% acetonitrile and 0.1% acetic acid. The elution was dried in a concentrator at 45°C. Peptides (20 µg) resuspended in a 100 mM triethylammonium bicarbonate solution were labeled using 100 µg of Thermo TMT10plex in a reaction volume of 25 µL for 1 hr. The labeling reaction was quenched with 8 µL of 5% hydroxylamine for 15 min. Labeled peptides were pooled, acidified to a pH of ~2 using drops of 10% trifluoroacetic acid, and desalted again with a Sep-Pak 50 mg C18 column as described above. TMT-labeled peptides were pre-fractionated using a Pierce High pH Reversed-Phase Peptide Fractionation Kit. Pre-fractionated peptides were dried using a concentrator and resuspended in 0.1% formic acid.

## Liquid chromatography coupled to tandem MS data acquisition

Pre-fractionated TMT-labeled peptides were analyzed on a Fusion Lumos mass spectrometer (Thermo Fisher Scientific) equipped with a Thermo EASY-nLC 1200 liquid chromatography (LC) system (Thermo Fisher Scientific). Peptides were separated by capillary reverse phase chromatography on a 25 cm column (75 µm inner diameter, packed with 1.6 µm C18 resin, AUR2-25075C18A, Ionopticks). Electrospray ionization voltage was set to 1550 V. Peptides were introduced into the Fusion Lumos mass spectrometer using a 180 min stepped linear gradient at a flow rate of 300 nL/min. The steps of the gradient were as follows: 6–33% buffer B (0.1% [vol/vol] formic acid in 80% acetonitrile) for 145 min, 33–45% buffer B for 15 min, 40–95% buffer B for 5 min, and 90% buffer B for 5 min. Column temperature was maintained at 50°C throughout the procedure. Xcalibur software (Thermo Fisher Scientific) was used for the data acquisition and the instrument was operated in data-dependent mode. Advanced peak detection was disabled. Survey scans were acquired in the Orbitrap mass analyzer (centroid mode) over the range of 380–1400 m/z with a mass resolution of 120,000 (at m/z 200). For MS1 (the survey scan), the normalized AGC target (%) was set at 250 and the maximum injection time was set to 100 ms. Selected ions were fragmented by the collision-induced dissociation (CID) method with normalized collision energies of 34 and the tandem mass spectra were acquired in the ion trap mass analyzer with the scan rate set to 'Rapid'. The isolation window was set to 0.7 m/z. For MS2 (the peptide fragmentation scan), the normalized AGC target (%) and the maximum injection time were set to 'standard' and 35 ms, respectively. Repeated sequencing of peptides was kept to a minimum by dynamic exclusion of the sequenced peptides for 30 s. The maximum duty cycle length was set to 3 s. Relative changes in peptide concentration were determined at the level of the MS3 (reporter ion fragmentation scan) by isolating and fragmenting the five most dominant MS2 ion peaks.

## Spectral searches

All raw files were searched using the Andromeda engine (*Thomason et al., 2007*) embedded in MaxQuant (version 2.3.1.0) (*Cox and Mann, 2008*). A reporter ion MS3 search was conducted using TMT10plex isobaric labels. Variable modifications included oxidation (M) and protein N-terminal acetylation. Carbamidomethylation of cysteines was a fixed modification. The number of modifications per peptide was capped at five. Digestion was set to tryptic (proline-blocked). Database search was conducted using the UniProt proteome Ecoli_UP000000625_83333. The minimum peptide length was seven amino acids. The false discovery rate was determined using a reverse decoy proteome (*Elias and Gygi, 2007*).

## Proteomics data analysis

Normalization and protein slope calculations were performed as described previously (*Lanz et al., 2022*). In brief, the relative signal difference between the TMT channels for each peptide was plotted against the normalized cell area for each of the bins of *E. coli* cells. For protein detection, we used a minimum of three unique peptide measurements per protein as a threshold. To derive the protein slope values shown in *Figure 5*, individual peptide measurements were consolidated into a protein

level measurement using Python's groupby.median. Peptides with the same amino acid sequence that were identified as different charge states or in different fractions were considered independent measurements. We summarized the size scaling behavior of individual proteins as a slope value derived from a regression. Each protein slope value was based on the behavior of all detected peptides.

For a given protein, we calculated the cell size-dependent slope as follows:

$y_i$ = relative signal in the $i$th TMT channel (median of all corresponding peptides in this channel)

$x_i$ = normalized cell size in the $i$th TMT channel (cell area for a given time point/mean cell area for the experiment)

The protein slope value was determined from a linear fit to the log-transformed data using the equation:

$$log_2\left(y\right) = Slope \cdot log_2\left(x\right)$$

Variables were log-transformed so that a slope of 1 corresponds to an increase in protein concentration that is proportional to the increase in cell volume, and a slope of –1 corresponds to 1/volume dilution. Pearson correlation coefficients and p-values were calculated using SciPy's pearsonr module in Python (*Virtanen et al., 2020*). The results were reproducible across the two replicates (*Figure 5—figure supplement 1*).

## Estimation of protein abundance using summed ion intensity

MS1-level peptide ion intensities from experiment #1 were used to estimate the relative protein abundance. For each protein, all peptide intensities were summed together using the 'Intensity' column of MaxQuant's evidence.txt file. To adjust for the fact that larger proteins produce more tryptic peptides, the summed ion intensity for each protein was divided by its amino acid sequence length.

## Calculation of distance from *oriC*

Gene coordinates were downloaded from EcoCyc database. The midpoint of each coding sequence was used to determine the circular distance (in base pairs) from *oriC* (base pair 3,925,859 of the total length of 4,641,652). A script was written to make three distance calculations. First, the direct distance (the midpoint of each gene minus 3,925,859 bp [*oriC*]) was determined. Next, the midpoint coordinate of each gene was subtracted by the total genome length (4,641,652 bp) and then subtracted again by the coordinate of *oriC* (3,925,859 bp). Finally, the midpoint coordinate of each gene was added to the total genome length (4,641,652 bp) and then subtracted by the coordinate of *oriC* (3,925,859 bp). The absolute values of these three calculations were then taken and the minimal values represent the circular distances between *oriC* and each gene.

## Sample preparation for RNA-seq experiments

CRISPRi *oriC* strain (SJ_XTL676) was grown in M9glyCAAT at 37°C and cells were collected 60, 120, 200, and 240 min after addition of 0.2% L-arabinose. Images of cells were acquired at each time point to determine cell area, as described above. For each time point, the aliquot was spun down, the supernatant was removed, and the pellet was flash-frozen. *E. coli* pellets were resuspended in ice-cold PBS and mixed with *C. crescentus* cells in approximately a 1-to-1 ratio based on $OD_{600}$. *C. crescentus* cells were originally included with the intent of using them as a spike-in reference. However, the fraction of final reads obtained from *C. crescentus* transcripts was inconsistent with the initial mixing ratios. In addition, without knowing the exact DNA concentration in each 1N-rich cell sample, this spike-in became purposeless and thus was ignored for analysis. Cells (50 μL) in PBS were mixed with 250 μL TRI Reagent (Zymo Research) and lysed by bead beating on a Fastprep 24 (MPbio). Cell debris were pelleted (14k rpm, 2 min) and the supernatant was recovered. RNA was then extracted using the direct-zol RNA microprep kit (Zymo Research). rRNA was then depleted using the NEBNext rRNA Depletion Kit for Bacteria (NEB, #E7850) and NEBNext Ultra II Directional RNA Library Prep Kit for Illumina (NEB, #E7760) was then used to prepare libraries for paired-end (2×150 bp) Illumina sequencing (Novogene).

## RNA-seq data analysis

A combined genome file of *E. coli* K-12 MG1655 and *C. crescentus* NA1000 with gene annotations was generated using a previously described approach (*Swaffer et al., 2023*). Read mapping statistics and genome browser tracks were generated using custom Python scripts. For quantification purposes, reads were aligned as 2×50 mers in transcriptome space against an index generated from the combined gene annotation model using Bowtie (version 1.0.1; settings: -e 200 -a -X 1000) (*Langmead et al., 2009*). Alignments were then quantified using eXpress (version 1.5.1) (*Roberts and Pachter, 2013*) as transcripts per million (TPM). TPM values were then recalculated after filtering for only *E. coli* genes, with TPM values below 1 were then removed from analysis.

The RNA slope (i.e. relative transcript concentration vs. cell area) was calculated for each gene as follows. TPM values were normalized to the mean of all values for that gene and then $\log_2$-transformed. The same normalization was applied to the cell area measurements for each condition. A linear model was fitted to the normalized $\log_2$ data and the slope of the linear regression was taken as the RNA slope.

## Estimation of the minimum genome dilution that limits transcription per nutrient condition

The cell area at which the total active RNAPs deviates from the WT scaling behavior was estimated to approximate the minimal genome dilution required to limit bulk transcription in cells growing in different media. Deviation in scaling behavior was found to occur at ~2 μm² during fast growth (M9glyCAAT, *Figure 3H*) or at ~1 μm² during slow growth (M9gly or M9ala, *Figure 6A and B*). Given such deviations in scaling behavior were observed in cells with a single chromosome copy (i.e. in 1N cells), the minimal limiting concentration is estimated to be, on average, 0.5 chromosome copies per μm² in M9glyCAAT or 1 chromosome copy per μm² in M9gly or M9ala. Based on estimates described in Appendix 1 (*Supplementary file 8*), the average chromosome concentration in WT cells is 0.85 chromosome copies per μm² for M9glyCAAT, 1.09 chromosome copies per μm² for M9gly, and 1.01 chromosome copies per μm² for M9ala. This suggests that the genome is already limiting or close to be limiting (within a few percents) in cells growing in M9gly and M9ala while its concentration is ~40% above the limiting concentration in cells growing M9glyCAAT.

## Mathematical model and simulations

Phenomenological model: We developed a phenomenological model to illustrate cell growth dynamics. In our models, the number of mRNAs ($X$) and the number of proteins ($Y$) in the cell are described by the differential equations:

$$\frac{dX}{dt} = r_1 \alpha_{RNAP}(X, Y) Y - \delta X$$

$$\frac{dY}{dt} = r_2 \alpha_{ribo}(X, Y) Y$$

In this model, the parameters $r_1$ and $r_2$ are coefficients defined as

$$r_1 = \#\text{ of mRNAs synthesized per minute} \times \frac{\#\text{ of RNAPs on mRNA synthesis}}{\#\text{ of RNAPs on total RNA synthesis}} \frac{\#\text{ of RNAPs in cell}}{\#\text{ of proteins in cell}}$$

$$r_2 = \#\text{ of protein synthesized per minute} \times \frac{\#\text{ of ribosomes in cell}}{\#\text{ of proteins in cell}}$$

Here, $\alpha_{RNAP}(X, Y)$ is the fraction of active RNAPs, $\alpha_{ribo}(X, Y)$ is the fraction of active ribosomes, and $\delta$ is the mRNA degradation rate. We assumed that the protein degradation is negligible because most bacterial proteins are stable (*Balakrishnan et al., 2022*; *Lin and Amir, 2018*). For the fraction of active RNAPs, we considered two different models (models A and B). In model A, the fraction of active RNAPs was modeled by $\alpha_{RNAP} = \frac{[Z]}{K_1 + [Z]}$. In model B, where we included different states of RNAP and promoter, the expression of $\alpha_{RNAP}$ is more complicated and is described in Appendix 3. Instead of considering a type of Michaelis-Menten (MM) reaction (*Klumpp and Hwa, 2008*), model B is based on the mass action law. For both models A and B, the ribosomes were modeled by $\alpha_{ribo} = \frac{[X]}{K_2 + [X]}$. We used $[Z]$ and $[X]$ to represent the genome and mRNA concentrations, respectively. We assumed that

the cell volume ($V$) and cell area ($A$) are proportional to the protein number $Y$, which is a good approximation (*Basan et al., 2015*; *Kubitschek et al., 1984*). While this assumption has not been verified in 1N cells, the concentration of ribosomes, which constitute most of the protein mass, were found to remain constant in these cells (*Figure 2A*). Namely, in our model, $V = cY$ and $A = c'Y$ with constants $c, c'$. Under balanced growth conditions, $[Z]$ was assumed to be constant. For the DNA-limited growth condition (as in 1N cells), we assumed that the genome does not duplicate and is kept at a single copy per cell ($Z(t) = 1$) while the cell volume continue to increase. The details for parameters used in this model are described in *Supplementary file 2*.

Numerical simulations: To simulate the model for balanced and DNA-limited growths, we first fixed a parameter set based on the literature (see *Supplementary file 2*, *Supplementary file 3*) and simulated *Equation 1* and *Equation 2* to reach the regime of balanced growth. Under balanced growth, we obtained the balanced growth vector $(X, Y)$, where $Y$ matches the initial protein number reported in the literature (see *Supplementary file 3*). Then, using the balanced growth vector as the initial condition, we simulated the balanced growth and DNA-limited growth using different assumptions of genome content $Z(t)$ (described in the previous paragraph). The simulation was performed using the MATLAB built-in function *ode23* with relative and absolute errors of the numerical values to be $10^{-4}$ and $10^{-5}$, respectively. For each growth condition, we obtained growth trajectories $(X(t), Y(t))$, and from $Y(t)$, we calculated the cell volume $V(t)$ and area $A(t)$ as described above. For model A, the fraction of active RNAPs was calculated by $\alpha_{RNAP} = \frac{[Z]}{K_1 + [Z]}$, while for model B, it is calculated using a more complicated formula that takes into account RNAP kinetics (see Appendix 3). For both models, the fraction of active ribosomes was calculated as $\alpha_{ribo} = \frac{[X]}{K_2 + [X]}$.

Parameter fitting: For either model A or B, we fitted our simulation results to all the experimental data including growth curves, the fraction of active RNAPs, and the fraction of active ribosomes for 1N and WT or multi-N cells (six datasets in total); see Appendix 2 for the initial parameter estimation and Appendix 4 for the optimization procedure details. The optimized parameter sets (see *Supplementary file 3*) were used for the numerical simulations shown in *Figure 5*. Note that the optimized parameter set remained within the realistic range for cellular physiology (see Appendix 4—figure 1 for comparison between original and optimized parameters).

## Acknowledgements

We thank Drs. Brun, Jun, Marczynski, Shapiro, and Weisshaar for sharing published strains. We are grateful to Dr. Lavis for the gift of JF549 and Dr. Badrinarayanan for valuable discussion. We would also like to thank the Jacobs-Wagner laboratory for support, discussion, and critical reading of the manuscript. CJW is an investigator of the Howard Hughes Medical Institute.

## Additional information

### Funding

| Funder | Grant reference number | Author |
|---|---|---|
| Howard Hughes Medical Institute | | Christine Jacobs-Wagner |

The funders had no role in study design, data collection and interpretation, or the decision to submit the work for publication.

### Author contributions

Jarno Mäkelä, Conceptualization, Resources, Data curation, Software, Formal analysis, Investigation, Visualization, Methodology, Writing – original draft, Writing – review and editing; Alexandros Papagiannakis, Resources, Data curation, Software, Formal analysis, Investigation, Visualization, Methodology, Writing – original draft, Writing – review and editing; Wei-Hsiang Lin, Software, Formal analysis, Investigation, Visualization, Methodology, Writing – review and editing, Shared co-authorship with Michael C. Lanz; Michael Charles Lanz, Resources, Data curation, Software, Formal analysis, Investigation, Visualization, Methodology, Writing – review and editing, shared co-authorship with Wei-Hsiang

Lin; Skye Glenn, Resources, Data curation, Formal analysis, Investigation, Visualization, Methodology, Writing – review and editing; Matthew Swaffer, Resources, Formal analysis, Investigation, Visualization, Methodology, Writing – review and editing; Georgi K Marinov, Software, Formal analysis, Methodology, Writing – review and editing; Jan M Skotheim, Project administration, Writing – review and editing; Christine Jacobs-Wagner, Conceptualization, Supervision, Funding acquisition, Investigation, Visualization, Writing – original draft, Project administration, Writing – review and editing

### Author ORCIDs
Jarno Mäkelä ⓘ https://orcid.org/0000-0003-1844-2619
Alexandros Papagiannakis ⓘ https://orcid.org/0000-0002-6363-804X
Wei-Hsiang Lin ⓘ https://orcid.org/0000-0001-8177-5892
Christine Jacobs-Wagner ⓘ https://orcid.org/0000-0003-0980-5334

Reviewer #1 (Public review): https://doi.org/10.7554/eLife.97465.3.sa1
Reviewer #2 (Public review): https://doi.org/10.7554/eLife.97465.3.sa2
Reviewer #3 (Public review): https://doi.org/10.7554/eLife.97465.3.sa3
Author response https://doi.org/10.7554/eLife.97465.3.sa4

---

## Additional files

### Supplementary files
• Supplementary file 1. Gene-specific protein slopes calculated from the tandem-mass-tag mass spectrometry measurements in growing 1N-rich or mutli-N cell populations.

• Supplementary file 2. Description of the model parameters.

• Supplementary file 3. Initial and optimized model parameters.

• Supplementary file 4. Gene-specific RNA slopes calculated using RNA sequencing in growing 1N-rich cell populations. The RNA slopes are also compared with the average protein slopes.

• Supplementary file 5. Comparison between transcriptome and proteome remodeling statistics (RNA and protein slopes, respectively) with previously published gene expression statistics (*Balakrishnan et al., 2022*), or gene essentiality data (*Gerdes et al., 2003*; *Goodall et al., 2018*; *Hashimoto et al., 2005*).

• Supplementary file 6. Strains used in this study. The abbreviations *kan*, *cat*, and *spec* refer to gene cassette insertions conferring resistance to kanamycin, chloramphenicol, and spectinomycin, respectively. These insertions are flanked by Flp site-specific recombination sites (*frt*) that allow the removal of the insertion using Flp recombinase from plasmid pCP20 (*Cherepanov and Wackernagel, 1995*).

• Supplementary file 7. Oligonucleotides used in this study.

• Supplementary file 8. Sizes, DNA concentrations, and growth rates of cells in different growth media.

• Supplementary file 9. mRNA and protein numbers per cells and bulk rates of transcription and translation. This table describes how kinetic constants were estimated from the literature. Parameters $X_{ini}$, $Y_{ini}$, $r_1$, $r_2$ were used in ordinary differential equation (ODE) simulations. We assumed exponential growth and used the relation $M_{ini} = M/(2log2)$, where $M_{ini}$ is the biomass of a newborn cell and $M$ is the average cellular biomass in the population (*Koch and Schaechter, 1962*). The bulk transcription rate $r_1$ was defined as $r_1 = \frac{mRNA\,synthesis\,rate}{of\,proteins \in the\,cell}$, and the bulk translation rate $r_2$ was defined as $r_2 = \frac{protein\,synthesis\,rate}{of\,proteins \in the\,cell}$. Values were estimated from a previous study (*Bremer and Dennis, 2008*). For our estimations, we assumed that the average protein length is 310 amino acids and that the average mRNA length is about 1 kb (*Ishihama et al., 2008*).

• Supplementary file 10. DNA and mRNA affinity constants ($K_1$, $K_2$). The active fraction of RNA polymerases (RNAPs) and ribosomes, $\alpha_{RNAP}$ and $\alpha_{ribo}$, are given by the formulae $\alpha_{RNAP} = \frac{[Z]}{K_1+[Z]}$ and $\alpha_{ribo} = \frac{[X]}{K_2+[X]}$, where $[Z]$ and $[X]$ are the DNA concentration and the mRNA concentration in the cells, respectively. To infer the parameters $K_1$, $K_2$, we used the values of $\alpha_{RNAP}$, $\alpha_{ribo}$, $[Z]$, and $[X]$ of wild-type cells determined in our study and back-calculated the values of $K_1$, $K_2$.

• Supplementary file 11. Estimation of the mRNA degradation rate. The mRNA degradation data

were obtained from an experimental study (*Balakrishnan et al., 2022*).

- Supplementary file 12. Comparison between the model formulation with constant RNA polymerase (RNAP) concentration (model A) and the more complex model with increasing RNAP concentration (model B).
- MDAR checklist

## Data availability

Microscopy data and analysis code are available on the Biostudies - BioImage server (S-BIAD1350) and the Jacobs-Wagner lab GitHub repository (copy archived at *Mäkelä et al., 2024*). The TMT-MS and RNA-seq data analysis and processing code is available here and here (copy archived at *Lanz, 2024* and *Marinov, 2023*). For this study, the following TMT-MS and RNA-seq data were generated and stored in *Supplementary file 1*, *Supplementary file 4*, *Supplementary file 5*, which are included as supplements in the publication. *Supplementary file 1* includes the calculated protein slopes from 1N-rich and multi-N cells, the normalized protein proportions, the summed ion intensities for each protein as well as the distance of its gene from the origin of replication (oriC) and the GO-term analysis. *Supplementary file 4* includes the calculated mRNA slopes, in comparison with the average protein slopes from 1N-rich cells. *Supplementary file 5* is an ensemble of the TMT-MS data, the RNA-seq data, and previously published datasets of essential genes (*Gerdes et al., 2003*; *Goodall et al., 2018*; *Hashimoto et al., 2005*) and gene expression-related data (*Balakrishnan et al., 2022*). All sequencing data associated with this study have been deposited to the GEO repository with the GSE261497 accession number. The mass spectrometry proteomics data have been deposited to the ProteomeXchange Consortium via the PRIDE partner repository with the dataset identifier PXD050093.

The following datasets were generated:

| Author(s) | Year | Dataset title | Dataset URL | Database and Identifier |
|---|---|---|---|---|
| Mäkelä J, Papagiannakis A, Lin WH, Lanz MC, Glenn S, Swaffer M, Marinov GK, Skotheim JK, Jacobs-Wagner C | 2024 | Genome concentration limits cell growth and modulates proteome composition in *Escherichia coli* | https://www.ncbi.nlm.nih.gov/geo/query/acc.cgi?acc=GSE261497 | NCBI Gene Expression Omnibus, GSE261497 |
| Mäkelä J, Papagiannakis A, Lin WH, Lanz MC, Glenn S, Swaffer M, Marinov GK, Skotheim JK, Jacobs-Wagner C | 2024 | Genome concentration limits cell growth and modulates proteome composition in *Escherichia coli* | https://www.ebi.ac.uk/pride/archive/projects/PXD050093 | PRIDE, PXD050093 |
| Mäkelä J, Papagiannakis A, Lin WH, Lanz MC, Glenn S, Swaffer M, Marinov GK, Skotheim JM, Jacobs-Wagner C | 2024 | Genome concentration limits cell growth and modulates proteome composition in *Escherichia coli* | https://doi.org/10.6019/S-BIAD1350 | Biostudies, 10.6019/S-BIAD1350 |

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

## Appendix 1

### Estimation of the average genome concentrations

#### Cell area, cell volume, and growth rate

The cell width $w$, cell length $L$, and cell area $A$ under different conditions were measured in our experiments. From these values, we estimated the cell volume by $V(L,w) = \left(\frac{4\pi}{3}\right)\left(\frac{w}{2}\right)^3 + \pi\left(\frac{w}{2}\right)^2(L-w)$, assuming that the *E. coli* cell poles are hemispheres. For different culture conditions, the normalized growth rate $\lambda = \frac{1}{A}\frac{dA}{dt}$ was estimated by tracking the cell area increase during time-lapse imaging, using the UNet segmentation algorithm (*Zhou et al., 2020*) with customized MATLAB scripts (see https://github.com/JacobsWagnerLab). The doubling time $\tau_{DB}$ is calculated by $\tau_{DB} = log2/\lambda$. All values are listed in *Supplementary file 8* below.

#### DNA concentration

To estimate the average DNA concentration for cells under balanced growth, we assumed that the cell volume grows exponentially during the cell cycle and that the DNA content follows a specific DNA replication pattern (see *Appendix 1—figure 1A*). We used published information on the pattern of a fluorescent fusion to the DNA replication marker SeqA (*Govers et al., 2024*) to extract information about DNA replication. If a cell cycle event (i.e. initiation of DNA replication) happened at the division cycle percentile $\theta$ (from 0% to 100% corresponding to cell birth and division, respectively), this value can be inferred from population statistics with a suitable cell cycle marker. A demograph (*Hocking et al., 2012*) can be constructed by sorting cells by length, allowing one to identify the demograph percentile $\phi$ (from 0% to 100%) of the cell cycle event (see *Appendix 1—figure 1B*). To adjust for the non-uniform age distribution in exponentially growing populations, a conversion formula between cell cycle percentile and demograph percentile was used, namely, $\theta = 1 - \frac{log(2-\phi)}{log(2)}$ (*Wold et al., 1994*).

To take into account the overlapping DNA replication rounds in cells growing in rich medium (M9glyCAAT), we defined $\theta_1$ and $\theta_2$ as the cell cycle percentiles for initiation and termination of DNA replication, respectively. These values were estimated from the fluorescent SeqA pattern on the demograph (see *Appendix 1—figure 1B*), where the corresponding demograph percentiles were estimated to be $\phi_1 = 50\%$ and $\phi_2 = 60\%$. Using the conversion formula, we obtained $\theta_1 = 41.5\%$ and $\theta_2 = 51.5\%$. In the case of non-overlapping DNA replication in nutrient-poorer media (M9gly and M9ala), we define $\theta_B$ and $\theta_{B+C}$ as the cell cycle percentiles for initiation and termination of DNA replication, respectively. These values were estimated from the fluorescent SeqA pattern in the demograph (see *Appendix 1—figure 1B*), and the corresponding demograph percentiles were estimated. For the M9gly condition, we have $\phi_B = 0\%$ and $\phi_{B+C} = 64\%$ and this corresponded to $\theta_B = 0\%$ and $\theta_{B+C} = 56\%$. For the M9ala condition, we have $\phi_B = 41\%$ and $\phi_{B+C} = 77\%$ and this corresponded to $\theta_B = 33\%$ and $\theta_{B+C} = 71\%$.

Using these values, we extracted the DNA content $Z(t)$ and cell volume $V(t)$ during the division cycle (see *Appendix 1—figure 1B*) and calculated the mean DNA concentration $[Z]_{avg}$ by averaging $Z(t)/V(t)$ across the division cycle.

**A**

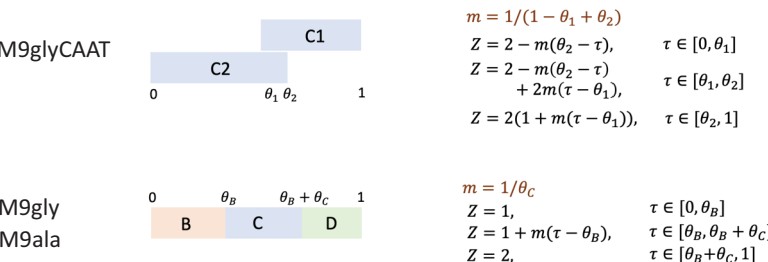

M9glyCAAT

$m = 1/(1 - \theta_1 + \theta_2)$
$Z = 2 - m(\theta_2 - \tau), \qquad \tau \in [0, \theta_1]$
$Z = 2 - m(\theta_2 - \tau) + 2m(\tau - \theta_1), \qquad \tau \in [\theta_1, \theta_2]$
$Z = 2(1 + m(\tau - \theta_1)), \qquad \tau \in [\theta_2, 1]$

M9gly
M9ala

$m = 1/\theta_C$
$Z = 1, \qquad \tau \in [0, \theta_B]$
$Z = 1 + m(\tau - \theta_B), \qquad \tau \in [\theta_B, \theta_B + \theta_C]$
$Z = 2, \qquad \tau \in [\theta_B + \theta_C, 1]$

**B**

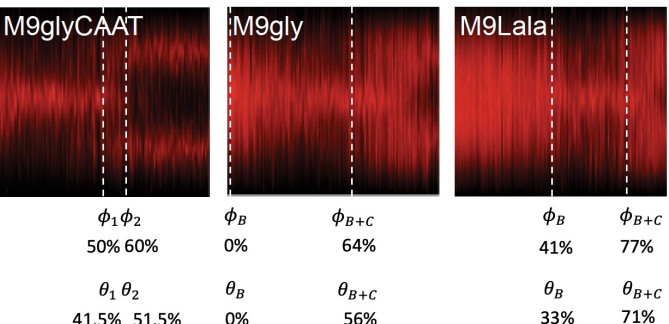

Adapted from Govers et. al. (2023)

| | M9glyCAAT | M9gly | M9Lala |
|---|---|---|---|
| | $\phi_1$ $\phi_2$ | $\phi_B$ $\phi_{B+C}$ | $\phi_B$ $\phi_{B+C}$ |
| | 50% 60% | 0% 64% | 41% 77% |
| | $\theta_1$ $\theta_2$ | $\theta_B$ $\theta_{B+C}$ | $\theta_B$ $\theta_{B+C}$ |
| | 41.5% 51.5% | 0% 56% | 33% 71% |

**C**

M9glyCAAT

mean genome concentration = 1.40 genome / $\mu m^3$

M9gly

mean genome concentration = 1.99 genome / $\mu m^3$

M9ala

mean genome concentration = 2.02 genome / $\mu m^3$

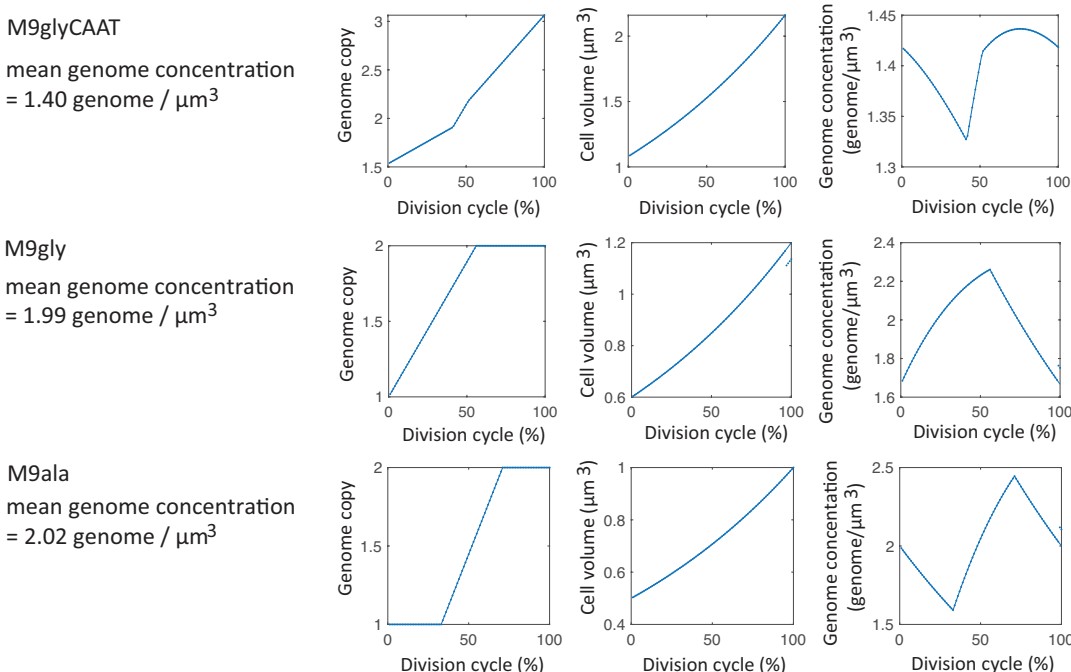

**Appendix 1—figure 1.** Estimation of DNA concentration. (**A**) DNA replication pattern for different medium conditions. (**B**) Extraction of the genome content, cell volume, and genome concentration along the division cycle. (**C**) Genome copy, cell volume, and genome concentration along the division cycle.

# Appendix 2

## Estimation of the model parameters

To simulate our ODE models (see *Equation 1* and *Equation 2* in the main text), we needed to estimate the kinetic constants during transcription and translation ($r_1, r_2, K_1, K_2, \delta$, see *Supplementary file 2* in the main text for details) and the ratio constants $c = \frac{cellvolume}{proteinnumber}$, $c' = \frac{cellarea}{proteinnumber}$. We also needed the values of protein and mRNA molecules in newborn cells ($X_{ini}, Y_{ini}$) as initial conditions for the simulations, as well as the DNA concentration of exponentially growing WT cells, $[Z]_{avg}$. For 1N cell, the DNA content was fixed as 1 copy per cell. In this section, we explain how these parameters were curated from the literature.

## Kinetic constants for transcription and translation

Our experiments were conducted in M9glyCAAT at 37°C. To estimate physiological parameters (mRNA and protein concentrations, etc.), we used datasets from the literature and interpolated using the doubling time (or growth rate) of our experimental condition as a reference (*Appendix 2—figure 1*). Three datasets (*Balakrishnan et al., 2022*; *Bremer et al., 2003*) were used for estimating the following kinetic constants:

(i) *Bremer and Dennis, 2008*: The authors curated physiological values of mRNA, protein, RNAP, and ribosome numbers in exponentially growing cells with doubling times $\tau = 20, 24, 30, 40, 60, 100\,\text{min}$ (all growing at 37°C using different carbon sources). In our experiments under M9glyCAAT, *E. coli* cells have a doubling time $\tau = 40\,\text{min}$. Hence, we used the values of $\tau = 40\,\text{min}$ from *Bremer and Dennis, 2008*, for our simulations.

(ii) *Bremer et al., 2003*: The authors calculated the fractions of RNAPs associated with rRNA (including *rrnP1* and *rrnP2*), mRNA (including constitutive and repressible genes), or under paused condition. These numbers were obtained for M9Glucose with casamino acid (with normalized growth rate 0.0277 min⁻¹) and M9Gly (with normalized growth rate 0.0115 min⁻¹), both at 37°C. To estimate the fraction of RNAPs engaged in mRNA synthesis, we used the formula as shown in *Appendix 2—figure 1G*, which we solved for the normalized growth rate of our growth condition.

(iii) *Balakrishnan et al., 2022*: The authors measured the mRNA degradation rates for normal and carbon-limited conditions, and showed that the mRNA degradation rate is not sensitive to growth rate. Therefore, we used the same mRNA degradation rate to model our medium condition (*Supplementary file 11*).

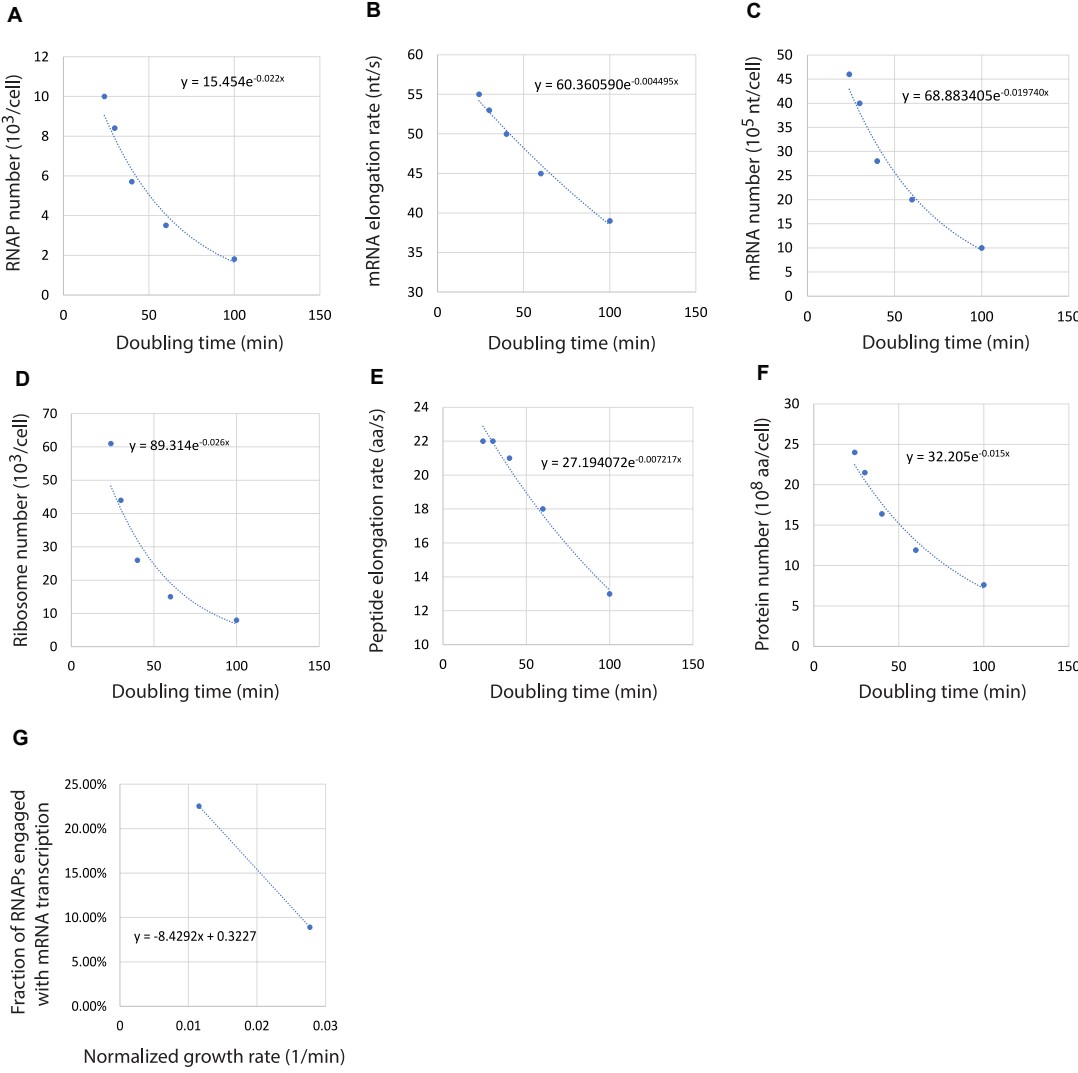

**Appendix 2—figure 1.** Interpolation and extrapolation of parameters. The parameters of the ODE models were calculated using the fitted formulae. The obtained values are summarized in **Supplementary file 9**.

## Appendix 3

### Model B, an alternative ODE model that includes the experimentally observed change in RNAP concentration in 1N cells

For the first ODE model A, we assumed that the active fraction of RNAPs follows MM type kinetics as $\alpha_{RNAP} = \frac{[Z]}{K_1 + [Z]}$. This model A only considered DNA concentration $[Z]$ but ignored the concentration of RNAP. In 1N cells, we experimentally found that the RNAP concentration increases with cell area (*Figure 3A*). An increasing RNAP concentration could potentially decrease $\alpha_{RNAP}$ due to an increased competition between RNAP molecules for DNA-binding sites (i.e. promoters). To take into account the increase in RNAP concentration in 1N cells, we considered a more sophisticated kinetic model B (see also *Supplementary file 12*). In this model , RNAP (denoted by $P$) is classified into three states:

1. Promoter-bound state ($P_p$)
2. Elongation state ($P_e$)
3. Free state ($P_f$)

The promoter (denoted by $Q$) is classified into two states:

1. RNAP-bounded state ($Q_b$)
2. Free state ($Q_f$)

The transition between these states is illustrated in the diagram below.

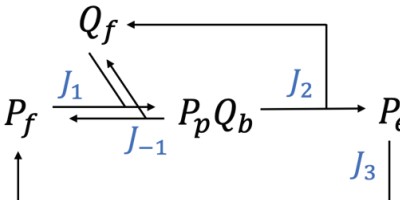

The fluxes between these states are:

$$J_1 = k_1[P_f][Q_f]$$
$$J_{-1} = k_{-1}[P_p] = k_{-1}[Q_b]$$
$$J_2 = k_2[P_p] = k_2[Q_b]$$
$$J_3 = k_3[P_e]$$

**Appendix 3—scheme 1.** Topology of the three RNA-polymerase states and their transition fluxes.

Here, $k_1$ and $k_{-1}$ are the ON and OFF constants between free RNAPs and unbounded promoters, respectively. $k_2$ is the rate constant for transcription initiation, while $k_3$ is the rate constant that is inversely proportional to the transcription time. Under steady state, we have

$$0 = \frac{d[P_f]}{dt} = (-J_1) + J_{-1} + J_3$$

$$0 = \frac{d[P_p]}{dt} = J_1 - J_{-1} - J_2$$

$$0 = \frac{d[P_e]}{dt} = J_2 - J_3$$

Solving the equations and let $[P] \equiv [P_f] + [P_p] + [P_e]$ and $[Q] \equiv [Q_f] + [Q_b]$, we get

$$[P] = [P_f]\left(1 + \frac{k_1}{k_{-1} + k_2}[Q_f] + \frac{k_1 k_2}{(k_{-1} + k_3)}[Q_f]\right) \tag{4-1}$$

$$[Q] = [Q_f]\left(1 + \frac{k_1}{k_{-1} + k_2}[P_f]\right) \tag{4-2}$$

We assume a separation of time scale such that $[P]$ (concentration of all RNAPs) and $[Q]$ (concentration of all promoters) are constant. Solving *Equations 4-1 and 4-2*, we obtain

$$[P_f] = \sqrt{C_1^2 + C_2} - C_1$$

$C_1 \equiv \frac{1 + A[Q] - B[P]}{2B}$ with ,

Note that the active fraction of RNAPs is $\alpha_{RNAP} \equiv 1 - \frac{[P_f]}{[P]}$, and the promoter occupancy is $\theta_{occ} \equiv \frac{[Q_b]}{[Q]}$. Therefore, the active fraction of RNAPs can be expressed as function of $[P]$ and $[Q]$, i.e.,

$$\alpha_{RNAP}\left([P],[Q]\right) = 1 - \left(\sqrt{f_1^2 + f_2} - f_1\right)$$

with $f_1 \equiv \frac{1 + A[Q] - B[P]}{2B[P]}$ with , $f_2 \equiv \frac{1}{B[P]}$, $A \equiv \frac{k_1}{k_{-1} + k_2}\left(1 + \frac{k_2}{k_3}\right)$, $B \equiv \frac{k_1}{k_{-1} + k_2}$

## Estimation of $[P]$, $[Q]$, and rate constants for model B

We needed to estimate the concentrations of RNAPs and promoters for cells in the M9glyCAA condition. From the constant $c = 0.283\ \mu m^3$/protein, the proteome fraction of RNAP $\theta_{RNAP} = 1.08 * 10^{-3}$, and the average cell volume $V = 1.5 \mu m^3$ (**Supplementary file 8**, **Supplementary file 9**), we obtain $[P] = 2.12 \times 10^3\left(\frac{\#}{\mu m^3}\right) = 3.39\ \mu M$.

The number of promoters in the cells was assumed to be $n_Q = 3000$ promoter (**Wang et al., 2013**). In M9glyCAA condition, the average genome copy is $[Z]_{avg} = 1.4\left(\frac{genome}{\mu m^3}\right)$ (**Supplementary file 8**). This corresponds to $[Q] = 4.2 \times 10^3\left(\frac{\#}{\mu m^3}\right) = 6.72\ \mu M$.

Under physiological condition, the concentrations of RNAPs and promoters are in the micromolar range. We obtained the following kinetic constants from the literature (**Bettridge et al., 2023**): $k_{-1} = 1.1\ s^{-1}$, $k_2 = 0.012 s^{-1}$, $k_3 = 0.0083 s^{-1}$. To estimate the remaining parameter $k_1$, we note that the WT cells in M9glyCAAT condition has $\alpha_{RNAP} = 0.5$ (**Supplementary file 10**) and this value corresponds to $k_1 = 7.5 \times 10^4 M^{-1} s^{-1}$ in model B.

## Estimation of the cell size dependence of RNAP concentration

From our experimental measurements, the RNAP concentration $[P]$ increases linearly with the cell area of 1N cells (orange line in **Figure 3A**). This can be described by the following phenomenological equation:

$$[P] = [P_0] + b\left(A - A_0\right)$$

If we let $1AU' = 10^5 AU$, then the slope $b$ is estimated to be $0.0786\ (AU'/\mu m^2)$.

To develop a biophysical model for this dependency, we rationalized that 1N cells can sense the dilution of DNA and upregulate their proteome fraction of RNAP ($\theta_{RNAP}^*$) accordingly. This suggests that $\theta_{RNAP}^*$ is a function of DNA concentration $[Z]$.

Since in WT condition $\theta_{RNAP}^*$ is only about 0.1% of the proteome (**Bremer and Dennis, 2008**), we assumed that the ratio between cell volume and total protein number ($c$) is a constant and that the ratio between cell area and the total protein number ($c'$) is also a constant. To apply these assumptions to 1N cells, we note that

$$[P] = \frac{\#RNAP}{V} = \frac{\#RNAP}{cY} = \frac{1}{c}\theta_{RNAP}^*$$

$$[P_0] = \frac{\#RNAP_0}{V_0} = \frac{\#RNAP_0}{cY_0} = \frac{1}{c}\theta_{RNAP}$$

where $\#RNAP_0$ represents the number of RNAPs in an average sized WT cell, and $[P_0], V_0, A_0, Y_0$ represent the RNAP concentration, the cell volume, the cell area, and the protein number, respectively. For 1N cells, the genome copy per cell is always 1, hence

$$A = \frac{c'}{c}V = \frac{c'}{c}\frac{1}{[Z]}$$

$$A_0 = \frac{c'}{c}V_0 = \frac{c'}{c}\frac{1}{[Z_0]}$$

Here, $[Z_0]$ is the average DNA concentration of 1N cells under normal cell size $V_0$ (before DNA dilution). For reference, we have $\theta_{RNAP} = 1.08 \times 10^{-3}$, $V_0 = 1.5\mu m^3$, $A_0 = 2.5\mu m^2$, $Z_0 = 0.67\mu m^{-3}$, $c = 0.28 \times 10^6 \mu m^3$, $c' = 0.47 \times 10^6 \mu m^2$ (see **Supplementary file 8**, **Supplementary file 9**). Substituting the above relation by the linear phenomenological equation $[P] = [P_0] + b(A - A_0)$, we have

$$\theta^*_{RNAP} = \theta_{RNAP} + bc(A - A_0) = \theta_{RNAP} + bc'\left(\frac{1}{[Z]} - \frac{1}{[Z_0]}\right)$$

To estimate the coefficient $b$ from experimental measurements, we needed to convert the RNAP concentration from AU' to (#/$\mu m^3$). We noticed that the RNAP concentration in WT cells is $\theta_{RNAP}c^{-1} = 4.15 \times 10^3$ (#/$\mu m^3$). This corresponds to 0.75 (AU') in the experimental measurements. Therefore, we have 1 (AU') = $5.53 \times 10^3$ (#/$\mu m^3$). From the experimental measurements, $b = 0.0786$ ($AU'/\mu m^3$), hence for 1N cells, we have $b = 430 \left(\frac{\#}{\mu m^5}\right)$. For multi-N cells, we simply assumed that $b = 0$ since based on our experimental measurements (**Figure 3A**, blue line), $[P]$ does not depend on cell area.

## Comparison between fraction of active RNAPs predicted by models A and B

To compare the two models, we first calculated the $\alpha_{RNAP}$ for both cases. Under physiological (WT-like cell sizes) conditions both the concentrations of RNAPs $[P]$ and promoters $[Q]$ are in micromolar range. In **Appendix 3—figure 1**, we see that both models give comparable landscape of $\alpha_{RNAP}$. Note that model A has no dependence on $[P]$, while model B shows some negative dependence on $[P]$.

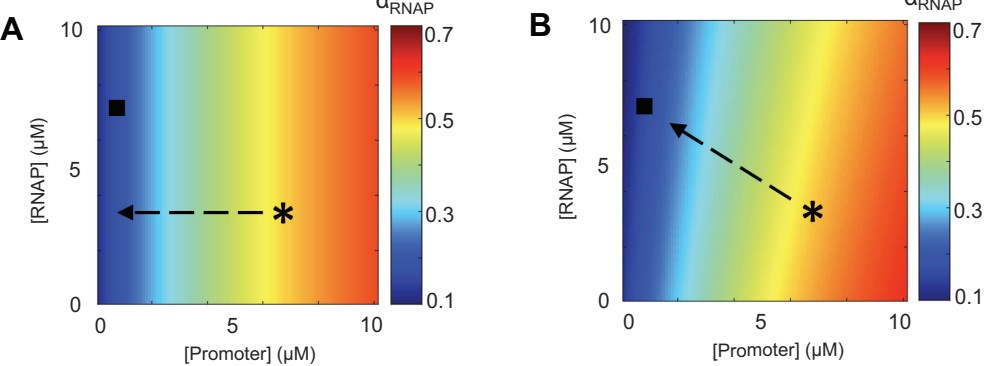

**Appendix 3—figure 1.** Comparing changes in active RNAP fraction between ODE models A and B. The two-dimensional colormaps show the values of active RNAP fraction ($\alpha_{RNAP}$) under different promoter and RNAP concentrations. (**A**) Using the formula of model A with M9glyCAAT parameters. (**B**) Using the formula of model B with M9glyCAAT parameters. The asterisk and filled square symbols indicate the promoter and RNAP concentration of wild-type cells and 1N cells (with 10$\mu m^2$ cell size), respectively. The dashed arrow indicates the trajectory of promoter and RNAP concentration under DNA-limited growth in the model.

## Appendix 4

### Fitting experimental data with the optimized ODE models

Parameter optimization for the ODE model

In Appendix 2, we curated the parameters from the literature and interpolated to match our experimental conditions. Based on the experimental methods used, the measured values may sometimes differ by 50% or more. In addition, we used interpolation and extrapolation to extract some values. Therefore, it is conceivable that our estimated values could deviate from the real physiological values. Since the goal of our mathematical model is to capture the general growth behavior of DNA limitation, we allowed the parameters to vary within a realistic range (see below) to improve the simultaneous fitting to six experimental datasets (growth rate, fraction of active RNAPs, and fraction of active ribosomes for both multi-N/WT and 1N cells).

We developed a scheme to obtain optimized parameters that fit the experimental data (as shown in *Figure 5A–C* and *Figure 5—figure supplement 1A–C*). For both model A and B, the optimization scheme focused on six parameters: $r_1$ (bulk mRNA synthesis rate), $r_2$ (bulk ribosome synthesis rate), $K_1$ (RNAP affinity to DNA), $K_2$ (ribosome affinity to mRNAs), $\delta$ (mRNA degradation rate), $c$ (ratio between cell volume and protein number). For the other parameters such as DNA concentration $[Z]$ and the number of protein and mRNA molecules for newborn cells $X_{ini}$, $Y_{ini}$, we fixed the values curated from the literature.

For the first-round optimization, we used the curated parameters from the literature (denoted as initial parameter set, see *Supplementary file 3* in the main text) and allowed parameters $(r_1, r_2, K_1, K_2, \delta, c)$ to randomly vary by ±50%. By repeating the same procedure 30,000 times, we explored the neighborhood of the initial parameter space. For each randomly varied parameter set, we simulated the ODE and compared the simulated results to our six experimental datasets. We calculated the error for each parameter set (see next section 'objective function for error minimization') and chose the parameter that yielded the smallest error. This chosen parameter set was then used for the next round of optimization.

For the $i$th round ($i = 2, \ldots$) of optimization, we used the optimized parameters from the previous round and randomly varied the parameters by ±5%. For each round, we repeated the same procedure 3000 times and chose the parameter set that gave the smallest error. The optimization iteration was stopped when the error reduction reached a plateau (relative error reduction was less than 1%), yielding to the final optimized parameter set. For model A, the optimization terminated in 9th, 8th, and 21st rounds, respectively. For model B, the optimization terminated in the third round. Comparison between initial and optimized parameter sets are shown in *Appendix 4—figure 1*.

Objective function for error minimization

In this section, we explain the details for calculating the error function. For each medium condition, six datasets were included in the optimization scheme:

1. Growth rate of multi-N (or WT) cells
2. Growth rate of 1N cells
3. Fraction of active RNAPs in multi-N (or WT) cells
4. Fraction of active RNAPs in 1N cells
5. Fraction of active ribosomes in multi-N (or WT) cells
6. Fraction of active ribosomes in 1N cells

Note that we used the multi-N cell dataset for M9glyCAAT condition and the WT cell datasets for the M9Gly and M9Ala conditions. Our goal was to choose the parameter set that minimized the error between ODE simulations and experiments. For each dataset $k = 1, \ldots, 6$, we defined the mean relative error by $E_k \equiv \sum_{j=1}^{N_k} \frac{\left| y_j^{data} - y_j^{simu} \right|}{y_j^{simu}}$, where $y_j^{data}$ is the $j$th datapoint in the experiment, $y_j^{simu}$ is the value predicted by the ODE model corresponded to this datapoint, and $N_k$ is the number of datapoints in the experiments. To take into account the six different datasets, we defined the objective function by $E \equiv \sum_{k=1}^{6} w_k E_k$, where $w_k$ is the weighting factors that balance the contribution between datasets. We set $w_k = 1$ for all datasets, since the multi-N and 1N datasets have comparable range of cell area in M9glyCAAT.

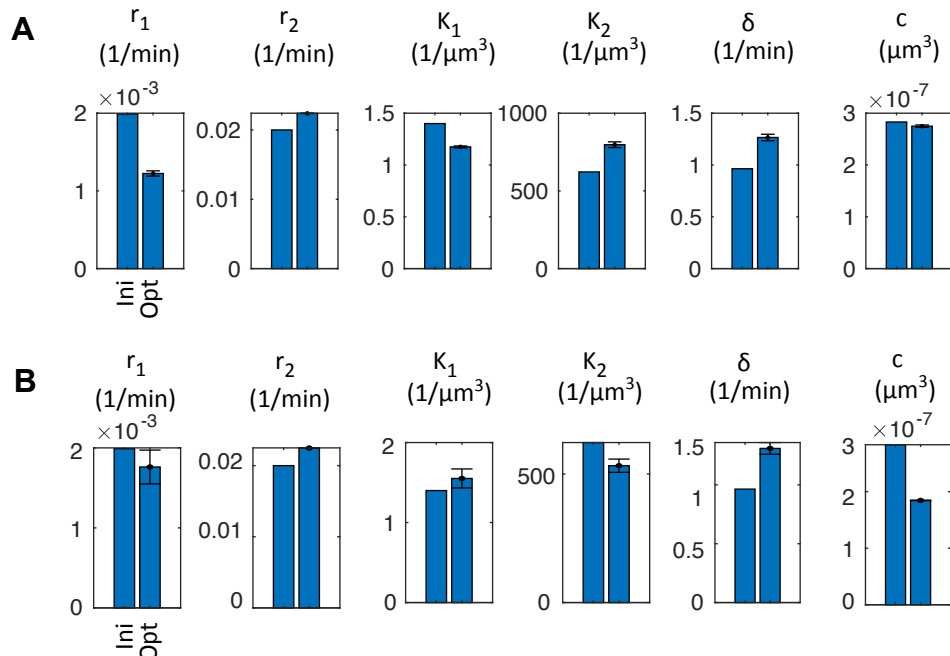

**Appendix 4—figure 1.** Comparison between initial (Ini) and optimized (Opt) parameters. (**A**) Parameters used in model A. (**B**) Parameters used in model B.

