## [Editor Report · eLife Assessment]

This **fundamental** work by Mäkelä et al. presents **compelling** experimental evidence supported by a theoretical model that the amount of chromosomal DNA can become limiting for the total rate of mRNA transcription and consequently protein production in the model bacterium *Escherichia coli*. The work is based on a mutant that allows inhibition of DNA replication while following growth at the single-cell level due to cell filamentation. The work significantly advances our understanding of growth and of the central dogma, and will be of considerable interest within both systems biology and microbial physiology.

---

## [Referee Report · Reviewer #1 (Public review)]

Summary:

The manuscript by Mäkelä et al. presents compelling experimental evidence that the amount of chromosomal DNA can become limiting for the total rate of mRNA transcription and consequently protein production in the model bacterium *Escherichia coli*. Specifically, the authors demonstrate that upon inhibition of DNA replication the rate of RNA transcription and the single-cell growth rate continuously decrease, the latter in direct proportion to the concentration of active ribosomes, as measured indirectly by single-particle tracking. The decrease of ribosomal activity with filamentation is likely caused by a decrease of the concentration of mRNAs, as suggested by an observed plateau of the total number of active RNA polymerases. These observations are compatible with the hypothesis that DNA limits the total rate of transcription and thus, indirectly, translation.

The authors also demonstrate that the decrease of RNAp activity is independent of two candidate stress response pathways, the SOS stress response and the stringent response, as well as an anti-sigma factor previously implicated in variations of RNAp activity upon variations of nutrient sources.

Remarkably, the reduction of growth rate is observed soon after the inhibition of DNA replication, suggesting that the amount of DNA in wild-type cells is tuned to provide just as much substrate for RNA polymerase as needed to saturate most ribosomes with mRNAs. While previous studies of bacterial growth have most often focused on ribosomes and metabolic proteins, this study provides important evidence that chromosomal DNA has a previously underestimated important and potentially rate-limiting role for growth.

Strengths:

This article links the growth of single cells to the amount of DNA, the number of active ribosomes and to the number of RNA polymerases, combining quantitative experiments with theory. The correlations observed during depletion of DNA, notably in M9gluCAA medium, are compelling and point towards a limiting role of DNA for transcription and subsequently for protein production soon after reduction of the amount of DNA in the cell. The article also contains a theoretical model of transcription-translation that contains a Michaelis-Menten type dependency of transcription on DNA availability and is fit to the data.

At a technical level, single-cell growth experiments and single-particle tracking experiments are well described, suggesting that different diffusive states of molecules represent different states of RNAp/ribosome activities, which reflect the reduction of growth.

Apart from correlations in DNA-deplete cells, the article also investigates the role of candidate stress response pathways for reduced transcription, demonstrating that neither the SOS nor the stringent response are responsible for the reduced rate of growth. Equally, the anti-sigma factor Rsd recently described for its role in controlling RNA polymerase activity in nutrient-poor growth media, seems also not involved according to mass-spec data. While other (unknown) pathways might still be involved in reducing the number of active RNA polymerases, the proposed hypothesis of the DNA substrate itself being limiting for the total rate of transcription is appealing.

Finally, the authors confirm the reduction of growth in the distant Caulobacter crescentus, which lacks overlapping rounds of replication and could thus have shown a different dependency on DNA concentration.

Weaknesses:

The study has no apparent weaknesses after review.

---

## [Referee Report · Reviewer #2 (Public review)]

In this work, the authors uncovered the effects of DNA dilution on *E. coli*, including a decrease in growth rate and a significant change in proteome composition. The authors demonstrated that the decline in growth rate is due to the reduction of active ribosomes and active RNA polymerases because of the limited DNA copy numbers. They further showed that the change in the DNA-to-volume ratio leads to concentration changes in almost 60% of proteins, and these changes mainly stem from the change in the mRNA levels.

Comments on revised version:

The authors have satisfyingly answered all of our questions.

---

## [Referee Report · Reviewer #3 (Public review)]

Mäkelä et al. here investigate genome concentration as a limiting factor on growth. Previous work has identified key roles for transcription (RNA polymerase) and translation (ribosomes) as limiting factors on growth, which enable an exponential increase in cell mass. While a potential limiting role of genome concentration under certain conditions has been explored theoretically, Mäkelä et al. here present direct evidence that when replication is inhibited, genome concentration emerges as a limiting factor.

A major strength of this paper is the diligent and compelling combination of experiment and modeling used to address this core question. The use of origin- and ftsZ-targeted CRISPRi is a very nice approach that enables dissection of the specific effects of limiting genome dosage in the context of a growing cytoplasm. While it might be expected that genome concentration eventually becomes a limiting factor, what is surprising and novel here is that this happens very rapidly, with growth transitioning even for cells within the normal length distribution for *E. coli*. Fundamentally, it demonstrates the fine balance of bacterial physiology, where the concentration of the genome itself (at least under rapid growth conditions) is no higher than it needs to be. A further surprising finding of this study is that susceptibility to this genome-limiting effect is felt differently by different genes, with unstable transcripts more affected and rRNA and many essential genes being more robust to it.

It should be noted that the authors do not identify a "smoking gun" - a gene or small number of genes that mediate the effects of genome concentration-dependent growth limitation. However, what they do achieve is to develop plausible criteria for identifying such a gene - through investigating essential genes that decrease in their abundance more rapidly than others.

Overall, this study provides a fundamental contribution to bacterial physiology by illuminating the relationship between DNA, mRNA, and protein in determining growth rate. While coarse-grained, the work invites exciting questions about how the composition of major cellular components is fine-tuned to a cell's needs and which specific gene products mediate this connection. The work also suggests the presence of buffering mechanisms that allow essential proteins such as RNA polymerase to be robust to fluctuations in genome concentration, which is an exciting area for future exploration. This work has implications not only for biotechnology, as the authors discuss, but potentially also for our understanding of how DNA-targeted antibiotics limit bacterial growth.

Comments on revised version:

Nothing left to add - the authors did a fantastic job addressing my points. In some ways doing so opened up even more interesting questions, but I happily accept that those are best left to future investigations.

---

## [Author Response]

The following is the authors’ response to the original reviews.

**Public Reviews:**

**Reviewer #1 (Public Review):**
Summary:The manuscript by Mäkelä et al. presents compelling experimental evidence that the amount of chromosomal DNA can become limiting for the total rate of mRNA transcription and consequently protein production in the model bacterium *Escherichia coli*. Specifically, the authors demonstrate that upon inhibition of DNA replication the single-cell growth rate continuously decreases, in direct proportion to the concentration of active ribosomes, as measured indirectly by single-particle tracking. The decrease of ribosomal activity with filamentation, in turn, is likely caused by a decrease of the concentration of mRNAs, as suggested by an observed plateau of the total number of active RNA polymerases. These observations are compatible with the hypothesis that DNA limits the total rate of transcription and thus translation. The authors also demonstrate that the decrease of RNAp activity is independent of two candidate stress response pathways, the SOS stress response and the stringent response, as well as an anti-sigma factor previously implicated in variations of RNAp activity upon variations of nutrient sources.Remarkably, the reduction of growth rate is observed soon after the inhibition of DNA replication, suggesting that the amount of DNA in wild-type cells is tuned to provide just as much substrate for RNA polymerase as needed to saturate most ribosomes with mRNAs. While previous studies of bacterial growth have most often focused on ribosomes and metabolic proteins, this study provides important evidence that chromosomal DNA has a previously underestimated important and potentially rate-limiting role for growth.

Thank you for the excellent summary of our work.

Strengths:This article links the growth of single cells to the amount of DNA, the number of active ribosomes and to the number of RNA polymerases, combining quantitative experiments with theory. The correlations observed during depletion of DNA, notably in M9gluCAA medium, are compelling and point towards a limiting role of DNA for transcription and subsequently for protein production soon after reduction of the amount of DNA in the cell. The article also contains a theoretical model of transcription-translation that contains a Michaelis-Menten type dependency of transcription on DNA availability and is fit to the data. While the model fits well with the continuous reduction of relative growth rate in rich medium (M9gluCAA), the behavior in minimal media without casamino acids is a bit less clear (see comments below).At a technical level, single-cell growth experiments and single-particle tracking experiments are well described, suggesting that different diffusive states of molecules represent different states of RNAp/ribosome activities, which reflect the reduction of growth. However, I still have a few points about the interpretation of the data and the measured fractions of active ribosomes (see below).Apart from correlations in DNA-deplete cells, the article also investigates the role of candidate stress response pathways for reduced transcription, demonstrating that neither the SOS nor the stringent response are responsible for the reduced rate of growth. Equally, the anti-sigma factor Rsd recently described for its role in controlling RNA polymerase activity in nutrient-poor growth media, seems also not involved according to mass-spec data. While other (unknown) pathways might still be involved in reducing the number of active RNA polymerases, the proposed hypothesis of the DNA substrate itself being limiting for the total rate of transcription is appealing.Finally, the authors confirm the reduction of growth in the distant Caulobacter crescentus, which lacks overlapping rounds of replication and could thus have shown a different dependency on DNA concentration.Weaknesses:There are a range of points that should be clarified or addressed, either by additional experiments/analyses or by explanations or clear disclaimers.First, the continuous reduction of growth rate upon arrest of DNA replication initiation observed in rich growth medium (M9gluCAA) is not equally observed in poor media. Instead, the relative growth rate is immediately/quickly reduced by about 10-20% and then maintained for long times, as if the arrest of replication initiation had an immediate effect but would then not lead to saturation of the DNA substrate. In particular, the long plateau of a constant relative growth rate in M9ala is difficult to reconcile with the model fit in Fig 4S2. Is it possible that DNA is not limiting in poor media (at least not for the cell sizes studied here) while replication arrest still elicits a reduction of growth rate in a different way? Might this have something to do with the naturally much higher oscillations of DNA concentration in minimal medium?

The reviewer is correct that there are interesting differences between nutrient-rich and -poor conditions. They were originally noted in the discussion, but we understand how our original presentation made it confusing. We reorganized the text and figures to better explain our results and interpretations. In the revised manuscript, the data related to the poor media are now presented separately (new Figure 6) from the data related to the rich medium (Figures 1-3). The total RNAP activity (abundance x active fraction) is significantly reduced in poor media (Figure 6A-B) similarly to rich medium (Figure 3H). Thus, DNA is limiting for transcription across conditions. However, the total ribosome activity in poor media (Figure 6C-D) and thus the growth rate (Figure 6EF) was less affected in comparison to rich media (Figure 2H and 1C). Our interpretation of these results is that while DNA is limiting for transcription in all tested nutrient conditions (as shown by the total active RNAP data), post-transcriptional buffering activities compensate for the reduction in transcription in poor media, thereby maintaining a better scaling of growth rates under DNA limitation.

The authors argue that DNA becomes limiting in the range of physiological cell sizes, in particular for M9glCAA (Fig. 1BC). It would be helpful to know by how much (fold-change) the DNA concentration is reduced below wild-type (or multi-N) levels at t=0 in Fig 1B and how DNA concentration decays with time or cell area, to get a sense by how many-fold DNA is essentially 'overexpressed/overprovided' in wild-type cells.

We now provide crude estimates in the Discussion section. The revised text reads: “Crude estimations suggest that ≤ 40% DNA dilution is sufficient to negatively affect transcription (total RNAP activity) in M9glyCAAT, whereas the same effect was observed after less than 10% dilution in nutrient-poor media (M9gly or M9ala) (see Materials and Methods).” We obtained these numbers based on calculations and estimates described in the Materials and Methods section and Appendix 1 (Appendix 1 – Table 1).

Fig. 2: The distribution of diffusion coefficients of RpsB is fit to Gaussians on the log scale. Is this based on a model or on previous work or simply an empirical fit to the data? An exact analytical model for the distribution of diffusion constants can be found in the tool anaDDA by Vink, ..., Hohlbein Biophys J 2020. Alternatively, distributions of displacements are expressed analytically in other tools (e.g., in SpotOn).

We use an empirical fit of Gaussian mixture model (GMM) of three states to the data and extract the fractions of molecules in each state. This avoids making too many assumptions on the underlying processes, e.g. a Markovian system with Brownian diffusion. The model in anaDDA (Vink et al.) is currently limited to two-transitioning states with a maximal step number of 8 steps per track for a computationally efficient solution (longer tracks are truncated). Using a short subset of the trajectories is less accurate than using the entire trajectory and because of this, we consider full tracks with at least 9 displacements. Meanwhile, Spot-On supports a three-state model but it is still based on a semi-analytical model with a pre-calculated library of parameters created by fitting of simulated data. Neither of these models considers the effect of cell confinement, which plays a major role in single-molecule diffusion in small-sized cells such as bacteria. For these reasons, we opted to use an empirical fit to the data. We note that the fractions of active ribosomes in WT cells, which we extracted from these diffusion measurements, are consistent with the range of estimates obtained by others using similar or different approaches (Forchhammer and Lindhal 1971; Mohapatra and Weisshaar, 2018; Sanamrad et al., 2014).

The estimated fraction of active ribosomes in wild-type cells shows a very strong reduction with decreasing growth rate (down from 75% to 30%), twice as strong as measured in bulk experiments (Dai et al Nat Microbiology 2016; decrease from 90% to 60% for the same growth rate range) and probably incompatible with measurements of growth rate, ribosome concentrations, and almost constant translation elongation rate in this regime of growth rates. Might the different diffusive fractions of RpsB not represent active/inactive ribosomes? See also the problem of quantification above. The authors should explain and compare their results to previous work.

We agree that our measured range is somewhat larger than the estimated range from Dai et al, 2016. However, they use different media, strains, and growth conditions. We also note that Dai et al did not make actual measurements of the active ribosome fraction. Instead, they calculate the “active ribosome equivalent” based on a model that includes growth rate, protein synthesis rate, RNA/protein abundance, and the total number of amino acids in all proteins in the cell. Importantly, our measurements show the same overall trend (a ~30% decrease) as Dai et al, 2016. Furthermore, our results are within the range of previous experimental estimates from ribosome profiling (Forchhammer and Lindhal 1971) or single-ribosome tracking (Mohapatra and Weisshaar, 2018; Sanamrad et al., 2014). We clarified this point in the revised manuscript.

To measure the reduction of mRNA transcripts in the cell, the authors rely on the fluorescent dye SYTO RNAselect. They argue that 70% of the dye signal represents mRNA. The argument is based on the previously observed reduction of the total signal by 70% upon treatment with rifampicin, an RNA polymerase inhibitor (Bakshi et al 2014). The idea here is presumably that mRNA should undergo rapid degradation upon rif treatment while rRNA or tRNA are stable. However, work from Hamouche et al. RNA (2021) 27:946 demonstrates that rifampicin treatment also leads to a rapid degradation of rRNA. Furthermore, the timescale of fluorescent-signal decay in the paper by Bakshi et al. (half life about 10min) is not compatible with the previously reported rapid decay of mRNA (24min) but rather compatible with the slower, still somewhat rapid, decay of rRNA reported by Hamouche et al.. A bulk method to measure total mRNA as in the cited Balakrishnan et al. (Science 2022) would thus be a preferred method to quantify mRNA. Alternatively, the authors could also test whether the mass contribution of total RNA remains constant, which would suggest that rRNA decay does not contribute to signal loss. However, since rRNA dominates total RNA, this measurement requires high accuracy. The authors might thus tone down their conclusions on mRNA concentration changes while still highlighting the compelling data on RNAp diffusion.

Thank you for bringing the Hamouche et al 2021 paper to our attention. To address this potential issue, we have performed fluorescence in situ hybridization (FISH) microscopy using a 16S rRNA probe (EUB338) to quantify rRNA concentration in 1N cells. We found that the rRNA signal only slightly decreases with cell size (i.e., genome dilution) compared to the RNASelect signal (e.g., a ~5% decrease for rRNA signal vs. 50% for RNASelect for a cell size range of 4 to 10 µm2). We have revised the text and added a figure to include the new rRNA FISH data (Figure 4). In addition, as a control, we validated our rRNA FISH method by comparing the intracellular concentration of 16S rRNA in poor vs. rich media (new Figure 4 – Figure supplement 3).

The proteomics experiments are a great addition to the single-cell studies, and the correlations between distance from ori and protein abundance is compelling. However, I was missing a different test, the authors might have already done but not put in the manuscript: If DNA is indeed limiting the initiation of transcription, genes that are already highly transcribed in non-perturbed conditions might saturate fastest upon replication inhibition, while genes rarely transcribed should have no problem to accommodate additional RNA polymerases. One might thus want to test, whether the (unperturbed) transcription initiation rate is a predictor of changes in protein composition. This is just a suggestion the authors may also ignore, but since it is an easy analysis, I chose to mention it here.

We did not find any correlation when we examined the potential relation between RNA slopes and mRNA abundance (from our first CRISPRi *oriC* time point) or the transcription initiation rate (from Balakrishnan et al., 2022, PMID: 36480614) across genes. These new plots are presented in Figure 7 – Figure supplement 2B. In contrast, we found a small but significant correlation between RNA slopes and mRNA decay rates (from Balakrishnan et al., 2022, PMID: 36480614), specifically for genes with short mRNA lifetimes (new Figure 7F). This effect is consistent with our model prediction (Figure 5 – Figure supplement 2).

Related to the proteomics, in l. 380 the authors write that the reduced expression close to the ori might reflect a gene-dosage compensatory mechanism. I don't understand this argument. Can the authors add a sentence to explain their hypothesis?

We apologize for the confusion. While performing additional analyses for the revisions, we realized that while the proteins encoded by genes close to *oriC* tend to display subscaling behavior, this is not true at the mRNA level (new Figure 7 – Figure supplement 3B). In light of this result, we no longer have a hypothesis for the observed negative correlation at the protein level (originally Figure 5D, now Figure 7 – Figure supplement 3A). The text was revised accordingly.

In Fig. 1E the authors show evidence that growth rate increases with cell length/area. While this is not a main point of the paper it might be cited by others in the future. There are two possible artifacts that could influence this experiment: (a) segmentation: an overestimation of the physical length of the cell based on phase-contrast images (e.g., 200 nm would cause a 10% error in the relative rate of 2 um cells, but not of longer cells). (b) timedependent changes of growth rate, e.g., due to change from liquid to solid or other perturbations. To test for the latter, one could measure growth rate as a function of time, restricting the analysis to short or long cells, or measuring growth rate for short/long cells at selected time points. For the former, I recommend comparison of phase-contrast segmentation with FM4-64-stained cell boundaries.

As the reviewer notes, the small increase in relative growth was just a minor observation that does not affect our story whether it is biologically meaningful or the result of a technical artefact. But we agree with the reviewer that others might cite it in future works and thus should be interpreted with caution.

An artefact associated with time-dependent changes (e.g. changing from liquid cultures to more solid agarose pads) is unlikely for two reasons. 1. We show that varying the time that cells spend on agarose pads relative to liquid cultures does not affect the cell size-dependent growth rate results (Figure 1 – supplement 5A). 2. We show that the growth rate is stable from the beginning of the time-lapse with no transient effects upon cell placement on agarose pads for imaging (Figure 1 – supplement 1). These results were described in the Methods section where they could easily be missed. We revised the text to discuss these controls more prominently in the Results section.

As for cell segmentation, we have run simulations and agree with the reviewer that a small overestimation of cell area (which is possible with any cell segmentation methods including ours) could lead to a small increase in relative growth with increasing cell areas (new Figure 1 – Figure supplement 3). Since the finding is not important to our story, we simply revised the text and added the simulation results to alert the readers to the possibility that the observation may be due to a small cell segmentation bias.

**Reviewer #2 (Public Review):**
In this work, the authors uncovered the effects of DNA dilution on *E. coli*, including a decrease in growth rate and a significant change in proteome composition. The authors demonstrated that the decline in growth rate is due to the reduction of active ribosomes and active RNA polymerases because of the limited DNA copy numbers. They further showed that the change in the DNA-to-volume ratio leads to concentration changes in almost 60% of proteins, and these changes mainly stem from the change in the mRNA levels.

Thank you for the support and accurate summary!

**Reviewer #3 (Public Review):**
Summary:Mäkelä et al. here investigate genome concentration as a limiting factor on growth.Previous work has identified key roles for transcription (RNA polymerase) and translation (ribosomes) as limiting factors on growth, which enable an exponential increase in cell mass. While a potential limiting role of genome concentration under certain conditions has been explored theoretically, Mäkelä et al. here present direct evidence that when replication is inhibited, genome concentration emerges as a limiting factor.Strengths:A major strength of this paper is the diligent and compelling combination of experiment and modeling used to address this core question. The use of origin- and ftsZ-targeted CRISPRi is a very nice approach that enables dissection of the specific effects of limiting genome dosage in the context of a growing cytoplasm. While it might be expected that genome concentration eventually becomes a limiting factor, what is surprising and novel here is that this happens very rapidly, with growth transitioning even for cells within the normal length distribution for *E. coli*. Fundamentally, it demonstrates the fine balance of bacterial physiology, where the concentration of the genome itself (at least under rapid growth conditions) is no higher than it needs to be.

Thank you!

Weaknesses:One limitation of the study is that genome concentration is largely treated as a single commodity. While this facilitates their modeling approach, one would expect that the growth phenotypes observed arise due to copy number limitation in a relatively small number of rate-limiting genes. The authors do report shifts in the composition of both the proteome and the transcriptome in response to replication inhibition, but while they report a positional effect of distance from the replication origin (reflecting loss of high-copy, origin-proximal genes), other factors shaping compositional shifts and their functional effects on growth are not extensively explored. This is particularly true for ribosomal RNA itself, which the authors assume to grow proportionately with protein. More generally, understanding which genes exert the greatest copy number-dependent influence on growth may aid both efforts to enhance (biotechnology) and inhibit (infection) bacterial growth.

We agree but feel that identifying the specific limiting genes is beyond the scope of the study. This said, we carried out additional experiments and analyses to address the reviewer’s comment and identify potential contributing factors and limiting gene candidates. First, we examined the intracellular concentration of 16S ribosomal RNA (rRNA) by rRNA FISH microscopy and found that it decays much slower than the bulk of mRNAs as measured using RNASelect staining (new Figure 4 and Figure 4 – Figure supplements 1 and 3). We found that the rRNA signal is far more stable in 1N cells than the RNASelect signal, the former decreasing by only ~5% versus ~50% for the later in response to the same range of genome dilution (Figure 4C). Second, we carried out new correlation analyses between our proteomic/transcriptomic datasets and published genome-wide datasets that report various variables under unperturbed conditions (e.g., mRNA abundance, mRNA degradation rates, fitness cost, transcription initiation rates, essentiality for viability); see new Figure 7E-G and Figure 7 – Figure supplement 2. In the process, we found that genes essential for viability tend, on average, to display superscaling behavior (Figure 7G). This suggests that cells have evolved mechanisms that prioritize expression of essential genes over nonessential ones during DNA-limited growth. Furthermore, this analysis identified a small number of essential genes that display strong negative RNA slopes (Figure 7C, Datasets 1 and 2), indicating that the concentration of their mRNA decreases rapidly relative to the rest of the transcriptome upon genome dilution. These essential genes with strong subscaling behavior are candidates for being growth-limiting.

The text and figures were revised to include these new results.

Overall, this study provides a fundamental contribution to bacterial physiology by illuminating the relationship between DNA, mRNA, and protein in determining growth rate. While coarse-grained, the work invites exciting questions about how the composition of major cellular components is fine-tuned to a cell's needs and which specific gene products mediate this connection. This work has implications not only for biotechnology, as the authors discuss, but potentially also for our understanding of how DNA-targeted antibiotics limit bacterial growth.

Thank you!

**Recommendations for the authors:**

**Reviewer #2 (Recommendations For The Authors):**
Below are my comments.(1) I noticed that a paper by Li et al. on biorxiv has found similar results as this work ("Scaling between DNA and cell size governs bacterial growth homeostasis and resource allocation," https://doi.org/10.1101/2021.11.12.468234), including the linear growth of *E. coli* when the DNA concentration is low. This relevant reference was not cited or discussed in the current manuscript.

We agree that authors should cite and discuss relevant peer-reviewed literature. But broadly speaking, we feel that extending this responsibility to all preprints (and by extension any online material) that have not been reviewed is a bit dangerous. It would effectively legitimize unreviewed claims and risk their propagation in future publications. We think that while imperfect, the peer-reviewing process still plays an important role.

Regarding the specific 2021 preprint that the reviewer pointed out, we think that the presented growth rate data are quite noisy and that the experiments lack a critical control (multi-N cells), making interpretation difficult. Their report that plasmid-borne expression is enhanced when DNA is severely diluted is certainly interesting and makes sense in light of our measurements that the activities, but not the concentrations, of RNA polymerases and ribosomes are reduced in 1N cells. However, we do not know why this preprint has not yet been published since 2021. There could be many possible reasons for this. Therefore, we feel that it is safer to limit our discussion to peer-reviewed literature.

(2) I think the kinetic Model B in the Appendix has been studied in previous works, such as Klump & Hwa, PNAS 2008, https://doi.org/10.1073/pnas.0804953105

Indeed, Klumpp & Hwa 2008 modeled the kinetics of RNA polymerase and promoter association prior to our study. But there is a difference between their model and ours. Their model is based on Michaelis Menten-type (MM) functions in which the RNAP is analogous to the “substrate” and the promoter to the “enzyme” in the MM equation. In contrast, our model uses functions based on the law of mass action (instead of MMtype of function). We have revised the text, included the Klumpp & Hwa 2008 reference, and revised the Materials & Methods section to clarify these points.

(3) On lines 284-285, if I understand correctly, the fractions of active RNAPs and active ribosomes are relative to the total protein number. It would be helpful if the authors could mention this explicitly to avoid confusion.

The fractions of active RNAPs and active ribosomes are expressed as the percentage of the total RNAPs and ribosomes. We have revised the text to be more explicit. Thank you.

(4) On line 835, I am not sure what the bulk transcription/translation rate means. I guess it is the maximum transcription/translation rate if all RNAPs/ribosomes are working according to Eq. (1,2). It would be helpful if the authors could explain the meaning of r_1 and r_2 more explicitly.

Our apology for the lack of clarity. We have added the following equations:×# of RNAPs on mRNA synthesis # of RNAPs on total RNA synthesis # of RNAPs in cell # of proteins in cell ×# of ribosomes in cell # of proteins in cell 

(5) Regarding the changes in protein concentrations due to genome dilution, a recent theoretical paper showed that it may come from the heterogeneity in promoter strengths (Wang & Lin, Nature Communications 2021).

In the Wang and Lin model, the heterogeneity in promoter strength predicts that the “mRNA production rate equivalent”, which is the mRNA abundance multiplied by the mRNA decay rate, will correlate the RNA slopes. However, we found these two variables to be uncorrelated see below, The Spearman correlation coefficient ρ was 0.02 with a p-value of 0.24, indicating non-significance (NS).

**Author response image 1. sa4fig1:** The mRNA production rate equivalent (mRNA abundance at the first time point after CRISPRi *oriC* induction multiplied by the mRNA degradation rate measured by Balakrishnan et al., 2022, PMID: 36480614, expressed in transcript counts per minute) does not correlate (Spearman correlation’s p-value = 0.24) with the RNA slope in 1N-rich cells. Data from 2570 genes are shown (grey markers, Gaussian kernel density estimation - KDE), and their binned statistics (mean +/- SEM, ~280 genes per bin, orange markers).

In addition, we found no significant correlation between RNA slopes and mRNA abundance or transcription initiation rate. These plots are now included in Figure 7E and Figure 7 –Figure supplement 2B. Thus, the promoter strength does not appear to be a predictor of the RNA (and protein) scaling behavior under DNA limitation.

**Reviewer #3 (Recommendations For The Authors):**
One general area that could be developed further is analysis of changes in the proteome/transcriptome composition, given that there may be specific clues here as to the phenotypic effects of genome concentration limitation. Specifically:• In Figure 5D, the authors demonstrate an effect of origin distance on sensitivity to replication inhibition, presumably as a copy number effect. However, the authors note that the effect was only slight and postulated a compensatory mechanism. Due to the stability of proteins, one should expect relatively small effects - even if synthesis of a protein stopped completely, its concentration would only decrease twofold with a doubling of cell area (slope = -1, if I'm interpreting things correctly). It would be helpful to display the same information shown in Figure 5D at the mRNA level, since I would anticipate that higher mRNA turnover rates mean that effects on transcription rate should be felt more rapidly.

We thank the reviewer for this suggestion. To our surprise, we found that there is no correlation between gene location relative to the origin and RNA slope across genes. This suggests that the observed correlation between gene location and protein slopes does not occur at the mRNA level. Given that we do not have an explanation for the underlying mechanism, we decided to present these data (the original data in Figure 5D and the new data for the RNA slope) in a supplementary figure (Figure 7 – Figure supplement 3).

• Related to this, did the authors see any other general trends? For example, do highly expressed genes hit saturation faster, making them more sensitive to limited genome concentration?

We found that the RNA slopes do not correlate with mRNA abundance or transcription initiation rates. However, they do correlate with mRNA decay. That is, short-lived mRNAs tend to have negative RNA slopes. The new analyses have been added as Figure 7E-F and Figure 7 – Figure supplement 2B. The text has been revised to incorporate this information.

• Presumably loss of growth is primarily driven by a subset of genes whose copy number becomes limiting. Previously, it has been reported that there is a wide variety among "essential" genes in their expression-fitness relationship - i.e. how much of a reduction in expression you need before growth is reduced (e.g. PMID 33080209). It would be interesting to explore the shifts in proteome/transcriptome composition to see whether any genes particularly affected by restricted genome concentration are also especially sensitive to reduced expression - overlap in these datasets may reveal which genes drive the loss of growth.

This is a very interesting idea – thank you! We did not find a correlation between the protein/RNA slope and the relative gene fitness as previously calculated (PMID 33080209), as shown below.

**Author response image 2. sa4fig2:** The relative fitness of each gene (data by Hawkins et al., 2020, PMID: 33080209, median fitness from the highest sgRNA activity bin) plotted versus the gene-specific RNA and protein slopes that we measured in 1Nrich cells after CRISPRi *oriC* induction. More than 260 essential genes are shown (262 RNA slopes and 270 protein slopes, grey markers), and their binned statistics (mean +/- SEM, 43-45 essential genes per bin, orange markers). The spearman correlations (ρ) with p-values above 10-3 are considered not significant (NS). In our analyses, we only considered correlations significant if they have a Spearman correlation p-value below 10-10.

However, while doing this suggested analysis, we noticed that the essential genes that were included in the forementioned study have RNA slopes above zero on average. This led us to compare the RNA slope distributions of essential genes relative to all genes (now included in Figure 7G). We found that they tend to display superscaling behavior (positive RNA slopes), suggesting the existence of regulatory mechanisms that prioritize the expression of essential genes over less important ones when genome concentration becomes limiting for growth. The text has been revised to include this new information.

Other suggestions:• In Figure 3 the authors report that total RNAP concentration increases with increasing cytoplasmic volume. This is in itself an interesting finding as it may imply a compensatory mechanism - can the authors offer an explanation for this?

We do not have a straightforward explanation. But we agree that it is very interesting and should be investigated in future studies given that this superscaling behavior is common among essential genes.

• The explanation of the modeling within the main text could be improved. Specifically, equations 1 and 2, as well as a discussion of models A and B (lines 290-301), do not explicitly relate DNA concentration to downstream effects. The authors provide the key information in Appendix 1, but for a general reader, it would be helpful to provide some intuition within the main text about how genome concentration influences transcription rate (i.e. via 𝛼RNAP).

We apologize for the lack of clarity. We have added information that hopefully improves clarity.